# Compact Bilinear Pooling via General Bilinear Projection

## Abstract

Many factorized bilinear pooling (FBiP) algorithms employ Hadamard product-based bilinear projection to learn appropriate projecting directions to reduce the dimension of bilinear features. However, in this paper, we reveal that the Hadamard product-based bilinear projection makes FBiP miss a lot of possible projecting directions, which will significantly harm the performance of outputted compact bilinear features, including compactness and effectiveness. To address this issue, we propose a general matrix-based bilinear projection based on the rank-$k$ matrix base decomposition, where the Hadamard-based bilinear projection and $\mathbf{Y} = \mathbf{U}^T \mathbf{X} \mathbf{V}$ are special cases of our proposed one. Thus, our proposed projection can be used improve the algorithms based on the two types of bilinear projections. Using the proposed bilinear projection, we design a novel low-rank factorized bilinear pooling (named RK-FBP), which considers the feasible projecting directions missed by the Hadamard product-based bilinear projection. Thus, our RK-FBP can generate better compact bilinear features. To leverage high-order information in local features, we nest several RK-FBP modules together to formulate a multi-linear pooling that outputs compact multi-linear features. At last, we conduct experiments on several fine-grained image tasks to evaluate our models, which show that our models achieve new state-of-the-art classification accuracy by the lowest dimension.

## 1    Introduction

Bilinear pooling (BiP) (Lin, 2015) and its variants (Li et al., 2017b; Lin & Maji, 2017; Wang et al., 2017) employ Kronecker product to yield expressive representations by mining the rich statistical information from a set of local features, and has attracted wide attentions in many applications, such as fine-grained image classification, visual question answering, etc. Although achieving excellent performance, the bilinear features suffer from two shortcomings: (1) the ability of BiP to boost the discriminant information between different classes also magnifies the intra-class variances of representations, which makes BiP easily encounter the burstiness problem (Gao et al., 2020; Zheng et al., 2019) and suffer from a performance deficit; (2) the Kronecker product exploited by BiP usually makes the bilinear features exceptionally high-dimensional, leading to an overfitted training of the succeeding tasks and a hefty computational load by increasing the memory storage. Thus, how to effectively solve the shortcomings of BiP is an important issue.

Several approaches have been proposed (Gao et al., 2016; Fukui et al., 2016; Yu et al., 2021; Li et al., 2017b; Kim et al., 2016) to solve the shortcomings of BiP. Among them, the factorized bilinear pooling (FBiP) methods (Li et al., 2017b; Kim et al., 2016; Amin et al., 2020; Yu et al., 2018; Gao et al., 2020) have been promising leads. The essence of FBiP performs a dimension reduction operation on bilinear features. It finds a linear projection to map bilinear features into a low-dimension space with their discriminant information among classes preserved using the least dimensions, and then employs $L_2$-normalization to project those low-dimension features on a hyper-sphere. (Bilinear pooling equals the non-linear projection determined by the polynomial kernel function $k(\mathbf{x}, \mathbf{y}) = (<\mathbf{x}, \mathbf{y}>)^2$ Gao et al. (2016), which makes bilinear features probably linear discriminant.) Thus, the information reflecting the large intra-class variances is abandoned because they do not help distinguish different classes. In this way, the burstiness and high dimension problems are solved simultaneously (Gao et al., 2020; Wei et al., 2018). This procedure is depicted in Figure 1. The sub-figure (a) shows a set of samples with a large variance. Sub-figure (c) is

Figure 1: (a): Original data $\{\mathbf{x}_i\}$; (b): The results projected by $\hat{\mathbf{x}}_i = [0.55, -0.45; -0.45, 0.55]\mathbf{x}_i$; (c): $L_2$-normalization on $\hat{\mathbf{x}}_i$: $\mathbf{z}_i = \hat{\mathbf{x}}_i/|\hat{\mathbf{x}}_i|_2$; (d): Element-wise "signed-square-root": $\tilde{\mathbf{x}}_i = sgn(\mathbf{x}_i)\sqrt{\mathbf{x}_i}$. Samples in regions 1,2,3,4 shrink to points $[1, 1]$, $[-1, 1]$,$[-1, -1]$ and $[1, -1]$, respectively. (e)(f): $L_2$-normalization on $\tilde{\mathbf{x}}_i$ and $\mathbf{x}_i$, respectively. Compared (e) with (f), we know the intra-class variances in region 2 and 4 are reduced but those in 1 and 3 are increased in (e). Thus, an accurate linear projection is better than "signed-square-root" strategy for solving the burstiness problem.

the result after the linear projection and $L_2$-normalization, the intra-class variance is reduced significantly. Actually, the information along $\mathbf{e}_1$ can be completely discarded, and the dimension of data becomes 1. Let us compare sub-figure (c) with (e), we can find FBiP outperforms the combination of the signed-square-root transformation and $L_2$-normalization . To achieve such a good performance, the key step is to find the accurate projecting direction $\mathbf{e}_2$, or the dimension and intra-variance reduction can not achieve successfully. Because of the extremely high dimension of bilinear features, it is not easy to find appropriate projecting directions by traditional linear projection. Thus, how to find a set of parameter-efficient model to accurately depict those appropriate directions is curial for solving the shortcomings of bilinear features.

Most FBiP approaches formulate the projection for dimension reduction as a Hadamard product-based bilinear projection (Kim et al., 2016; Gao et al., 2020; Li et al., 2017b): $\mathbf{f} = \mathbf{P}^T(\mathbf{U}^T\mathbf{x}_s \circ \mathbf{V}^T\mathbf{y}_t)$ where $\mathbf{U}$ and $\mathbf{V}$ are two learnable variables, respectively, and $\mathbf{P}$ is a variable defined various from algorithms. Each dimension of $\mathbf{f}$ can be seen as a linear combination of dimensions of the Hadamard product $\mathbf{z} = (\mathbf{U}^T\mathbf{x}_s \circ \mathbf{V}^T\mathbf{y}_t)$. Consider each dimension of $\mathbf{z}$ shown in Figure 2. The Hadamard product only considers the values $\{(\mathbf{u}_i^T\mathbf{x}_s)(\mathbf{y}_t^T\mathbf{v}_j)|i = j)\}$ and ignores values $\{(\mathbf{u}_i^T\mathbf{x}_s)(\mathbf{y}_t^T\mathbf{v}_j)|i \neq j\}$ which are considered by Kronecker

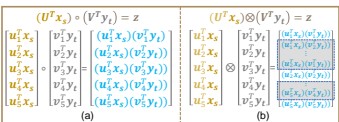

Figure 2: Two ways to connect dimensions of $\mathbf{U}^T\mathbf{x}_s$ and $\mathbf{V}^T\mathbf{y}_t$. (a) is Hadamard product; (b) is Kronecker product.

product. In this paper, we prove that those ignored values are important for dimension reduction, because they are coefficients of bilinear features on feasible matrix projecting directions $\{\mathbf{v}_i\mathbf{u}_j^T|i \neq j\}$. Thus, missing them will lead to inaccurate projecting directions of the dimension reduction, which inevitably affects the overcoming for shortcomings of BiP greatly. Consequently, the effectiveness and compactness of FBiP are seriously harmed.

In this paper, from the perspective of finding accurate and parameter-efficient matrix projecting directions, we analyze the decomposition on bases of a matrix space, then propose a general bilinear projection based on decomposed rank-$k$ matrix bases. Because of the solid mathematical foundation of the proposed bilinear projection, it can be seen as a baseline to analyze current FBiP. Employing our novel bilinear projection, we formulate a new FBiP model without missing possible projecting directions. The contributions are listed as follows:

(1) We make a detailed analysis to demonstrate why the traditional FBiP tends to miss a lot of feasible projecting directions. Based on our analysis, we propose a general bilinear projection that calculates the coefficients of matrix data on a set of complete decomposed rank-$k$ matrix bases.

(2) Based on the proposed general bilinear projection, we design a new FBiP method named rank-$k$ factorized bilinear pooling (RK-FBP). Because of the capability to learn accurate projecting directions, the calculated bilinear features are highly compact and effective. Utilizing this property, we nest several RK-FBP modules to calculate multi-linear features that are still compact and effective.

(3) We conduct experiments on several challenging image classification datasets to demonstrate the effectiveness of the proposed RK-FBP. Compared with state-of-the-art BiP methods, our model can output extremely compact bilinear features (the dimension is $512$) and achieve comparable or better classification accuracy.

**Notation.** Throughout this paper, $Tr(\cdot)$ denotes the trace of a matrix. $vec(\cdot)$ is the matrix vectorization (i.e., reshaping a matrix to a vector by stacking its columns on top of one another). $rank(\cdot)$ represents the rank of a matrix. The $ij$-th element of the matrix $\mathbf{A} \in \mathbb{R}^{m \times n}$ is represented by the symbol $(\mathbf{A})_{ij}$, and the $i$-th element of $\mathbf{x} \in \mathbb{R}^d$ is represented by $(\mathbf{x})_i$.

## 2 PRELIMINARY

### 2.1 DIMENSION REDUCTION ON BILINEAR FEATURES

Given a set of training samples, among which each instance yields two groups of local features denoted by $\mathcal{X}_f = \{\mathbf{x}_s \in \mathbb{R}^m\}_{s=1}^p$ and $\mathcal{Y}_f = \{\mathbf{y}_t \in \mathbb{R}^n\}_{t=1}^q$ (in some cases, $\mathcal{Y}_f = \mathcal{X}_f$ (Lin, 2015; Kong & Fowlkes, 2017)). Bilinear pooling (BiP) integrates those two groups of local features into an expressive representation by the following operation.

$$\mathbf{X} = \sum_{(s,t) \in \mathcal{S}} \mathbf{x}_s \mathbf{y}_t^T \in \mathbb{R}^{m \times n} \tag{1}$$

where $\mathbf{X}$ is the bilinear feature, and $\mathcal{S}$ is the pair set of local features. For convenience, we write bilinear feature as $\mathbf{X} = \mathbf{x}_s \mathbf{y}_t^T$ in the following content by ignoring the summation symbol. Because most deep neural networks satisfy $|\mathcal{S}| < \min\{m, n\}$, the bilinear feature $\mathbf{X}$ is a low-rank matrix. Such property can help to design parameter-efficient dimension reduction algorithms.

Factorized bilinear pooling (FBiP) reduces $\mathbf{X}$ to a $h$-dimensional vector by linear projection $\mathbf{f} = [Tr(\mathbf{X}\mathbf{W}_1^T), \cdots, Tr(\mathbf{X}\mathbf{W}_h^T)]$ where $\mathbf{W}_r \in \mathbb{R}^{m \times n}$ is the $r$-th low-rank matrix projecting direction. For parameter-efficiency, each $\mathbf{W}_r$ is decomposed into small matrices in various ways, which leads to different FBiP algorithms.

For example, by decomposing $\mathbf{W}_r$ as $\mathbf{W}_r = \mathbf{U}_r \mathbf{V}_r^T$ where $\mathbf{U}_r \in \mathbb{R}^{m \times k}$ and $\mathbf{V}_r \in \mathbb{R}^{n \times k}$, $k$ is the rank of $\mathbf{W}_r$. The $r$-th dimension of $\mathbf{f}$ can be rewritten as follows.

$$(\mathbf{f})_r = Tr\left(\mathbf{x}_s \mathbf{y}_t^T \mathbf{V}_r \mathbf{U}_r^T\right) = \mathbb{1}^T (\mathbf{U}_r^T \mathbf{x}_s \circ \mathbf{V}_r^T \mathbf{y}_t) \tag{2}$$

where $\circ$ is Hadamard product and $\mathbb{1}$ is a vector with all elements equaling 1. The Eq.(2) is adopted by FBC (Gao et al., 2020) and LowFER (Amin et al., 2020).

Besides, some FBiP algorithms (Kong & Fowlkes, 2017; Wei et al., 2018) let local feature sets $\mathcal{X} = \mathcal{Y}$ and assume $\mathbf{W}_r$ be a symmetric low-rank matrix. The $\mathbf{W}_r$ is decomposed as $\mathbf{W}_r = \mathbf{U}_r^+ (\mathbf{U}_r^+)^T - \mathbf{U}_r^- (\mathbf{U}_r^-)^T$ where $\mathbf{U}_r^+$ and $\mathbf{U}_r^-$ are learned smaller matrices. Thus, let $\mathbf{U}_r = [\mathbf{U}_r^+, \mathbf{U}_r^-]$, $r$-th dimension also can be rewritten as a Hadamard product-based projection:

$$(\mathbf{f})_r = Tr\left(\mathbf{x}_s \mathbf{y}_t^T \mathbf{U}_r^+ (\mathbf{U}_r^+)^T\right) - Tr\left(\mathbf{x}_s \mathbf{y}_t^T \mathbf{U}_r^- (\mathbf{U}_r^-)^T\right) = [\mathbb{1}^T, -\mathbb{1}^T](\mathbf{U}_r^T \mathbf{x}_s \circ \mathbf{U}_r^T \mathbf{y}_t) \tag{3}$$

Some algorithms (Kim et al., 2016; Yu et al., 2018; Kim et al., 2018) replace $\mathbb{1}$ in Eq.(2) by a learnable vector $\mathbf{p}_r$ and transform Eq.(2) to a more general formulation presented as follows.

$$\mathbf{f} = \mathbf{P}^T \left(\mathbf{U}^T \mathbf{x}_s \circ \mathbf{V}^T \mathbf{y}_t\right) \tag{4}$$

where $\mathbf{P} = [\mathbf{p}_1, \cdots, \mathbf{p}_h] \in \mathbb{R}^{l \times h}$, $\mathbf{U} \in \mathbb{R}^{m \times l}$, and $\mathbf{V} \in \mathbb{R}^{n \times l}$ are learnable matrices, respectively. Eq.(4) is a general formulation of Eq.(2) and Eq.(3), mathematically, so we can only focus our analysis on Eq.(4).

### 2.2 SHORTCOMINGS OF TRADITIONAL FACTORIZED LOW-RANK BILINEAR POOLING

The linear dimension reduction finds the least projecting directions to mapping samples into the low-dimensional space with their discriminant information being preserved. Thus, the projecting directions should satisfy the following criteria: *(1) they preserve discriminant information of samples well, otherwise the classification performance will be poor; (2) those projecting directions are linear independent, or the calculated low-dimensional embeddings can be reduced further.*

Next, we demonstrate that the bilinear projection in Eq.(4) can not find suitable projecting directions and harms the compactness and effectiveness of the compact bilinear features. Before doing this, we first introduce a theorem presented as follows.

**Theorem 1**. *Suppose $\mathcal{U} = \{\boldsymbol{u}_p \in \mathbb{R}^{m \times 1}\}_{p=1}^{l_1}$ and $\mathcal{V} = \{\boldsymbol{v}_q \in \mathbb{R}^{n \times 1}\}_{q=1}^{l_2}$ are two groups of linear independent vectors in spaces $\mathbb{R}^{m \times 1}$ and $\mathbb{R}^{n \times 1}$, respectively. If the vector set $\mathcal{W} = \{vec(\boldsymbol{W}_i) \in \mathbb{R}^{mn \times 1}\}_{i=1}^{l_1 l_2}$ is constructed by $\boldsymbol{W}_i = \boldsymbol{u}_p \boldsymbol{v}_q^T$ where $i = l_1(q-1) + p$, then $\mathcal{W}$ is a set of linear independent vectors.*

The Proof is attached in the Appendix. If $l_1 = m$ and $l_2 = n$, $\mathcal{W}$ is a complete bases of $\mathbb{R}^{mn \times 1}$. The $r$-th element of the low-dimensional feature $\mathbf{f}$ presented in Eq.(4) can be rewritten as

$$(\mathbf{f})_r = \left( \sum_{j=1}^{l} (\mathbf{p}_r)_j vec(\mathbf{u}_j \mathbf{v}_j^T) \right)^T vec(\mathbf{x}_s \mathbf{y}_t^T) \tag{5}$$

where $(\mathbf{p}_r)_j$ is the $j$-th element in the $r$-th column of $\mathbf{P}$. As seen from Eq.(5), $(\mathbf{f})_r$ is the coefficient of the bilinear feature $vec(\mathbf{x}_s \mathbf{y}_t^T)$ projected on the projecting direction $\sum_{j=1}^{l} (\mathbf{p}_r)_j vec(\mathbf{u}_j \mathbf{v}_j^T)$ which is a vector in the linear space spanned by vectors $\mathcal{B} = \{vec(\mathbf{u}_1 \mathbf{v}_1^T), vec(\mathbf{u}_2 \mathbf{v}_2^T), \cdots, vec(\mathbf{u}_l \mathbf{v}_l^T)\}$.

Then, we prove that the linear combinations of vectors in $\mathcal{B}$ can not express all the possible projecting directions.

Let us start from the case $l \leq \max\{m, n\}$. Because learning a low-rank matrix is a challenging problem in the machine learning community(Candès et al., 2011; Liu et al., 2012; Wright et al., 2009), $\mathbf{U}$ and $\mathbf{V}$ in Eq.(4) are much likely to be full rank matrices because of no additional constraints on them. Thus, columns in $\mathbf{U}$ (and $\mathbf{V}$) are linear independent. Figure 3 gives experimental Proof to support this assumption. Because the columns of $\mathbf{U}$ trained by the FBiP model of Eq.(4) are nearly vertical, $\mathbf{U}$ is a full rank matrix.

According to Theorem 1, columns of $\mathbf{U}$ and $\mathbf{V}$ can generate a set of linear independent vectors $\mathcal{W}$. Obviously, $\mathcal{B}$ is a small subset of $\mathcal{W}$. According to criterion (2), not only vectors in the space spanned by $\mathcal{B}$ but also vectors spanned by $\mathcal{W} - \mathcal{B}$ can generate the possible projecting directions for reducing dimensions of samples in $\mathbb{R}^{mn \times 1}$.

However, because Eq.(4) only calculates coefficients of the bilinear feature projected on directions spanned by $\mathcal{B}$, it misses the projecting directions generated by $\mathcal{W} - \mathcal{B}$. Considering the number of vectors in $\mathcal{W} - \mathcal{B}$ and $\mathcal{B}$ are $l^2 - l$ and $l$, respectively, more than $99\%$ feasible projecting directions are missed if $\max\{m, n\} > l > 100$.

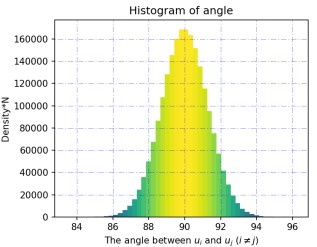

Worse, such a serious issue can not be alleviated by increasing $l$. Without loss of generality, we let $l > m = n$ and the first $m$ columns in $\mathbf{U}$ and $\mathbf{V}$ be linear independent. Thus, the last $l - m$ columns in $\mathbf{U}$ (or $\mathbf{V}$) can be represented by the first $m$ columns. To be specific, the $r$-th ($r > m$) columns in $\mathbf{U}$ and $\mathbf{V}$ are $\mathbf{u}_r = \sum_{p=1}^{m} a_{rp} \mathbf{u}_p$ and $\mathbf{v}_r = \sum_{q=1}^{m} b_{rq} \mathbf{v}_q$, respectively, where $a_{rp}$ and $b_{rq}$ are auxiliary parameters introduced for easy reading. Obviously, $\mathbf{u}_r$ and $\mathbf{v}_r$ can generate a projecting direction $vec(\mathbf{u}_r \mathbf{v}_r^T) = \sum_{p=1}^{m} \sum_{q=1}^{m} (a_{rp} b_{rq}) vec(\mathbf{u}_p \mathbf{v}_q^T)$. Let us list those generated projecting vectors of

Figure 3: The angles between columns of $\mathbf{U}(2048 \times 2048)$ trained by Eq.(4). Because the angle range is about $[85°, 95°]$, the columns are nearly vertical to each other.

different $r$ into a projecting matrix, i.e., $\hat{\mathbf{L}} = [vec(\mathbf{u}_{m+1} \mathbf{v}_{m+1}^T), \cdots, vec(\mathbf{u}_l \mathbf{v}_l^T)]$.

According to Theorem 1, $\{vec(\mathbf{u}_p \mathbf{v}_q^T)\}_{p=1,q=1}^{m,n}$ is a complete base set in $\mathbb{R}^{mn \times 1}$. So $vec(\mathbf{u}_r \mathbf{v}_r^T)$ can be any $mn$-dimensional vector by giving auxiliary parameters $\{a_{rp} b_{rq}\}_{p=1,q=1}^{m,n}$ suitable values. However, because those auxiliary parameters are implicit, we can not train them as other parameters of our model. Thus, the learnable projecting directions can be any possible ones in the solution space. The worst case is that $\hat{\mathbf{L}}$ is a rank-1 matrix, which means most of its columns do not preserve the discriminant information of samples. Thus, we can transform $\hat{\mathbf{L}}$ to one column without performance reduction. It implies that the missed projecting directions can not be found by increasing the value of $l$. Thus, FBiP can not find suitable projecting directions. Of course, auxiliary parameters can let $\hat{\mathbf{L}}$ be a full rank matrix. In this case, the projecting directions will be in the space spanned by the missed directions $\mathcal{W} - \mathcal{B}$. The performance of deep learning crucially depends on tricks in stochastic gradient descent strategies, such as 'momentum' and 'decay'. Because those auxiliary parameters can not be improved by those tricks, their values may not as good as desired, which probably make the learned projecting directions unsuitable. Thus, the performance of FBiP will

be poor. To alleviate this issue, FBiP should adopt more projection directions to capture enough discriminant information. Nevertheless, due to keeping large intra-class variances, the performance of those unsuitable solutions has an upper bound less than that of the suitable projecting directions. Consequently, the effectiveness and compactness of outputted bilinear features are harmed.

For the case $\mathbf{U} = \mathbf{V}$, the following Corollary can demonstrate Eq.(4) also misses a lot of feasible projecting matrices. The analysis can be made in the same way mentioned above, needing to replace Theorem 1 with the Corollary. Thus, we do not present the analysis in our paper.

**Corollary.** *Given a set of linear independent vectors $\{\boldsymbol{u}_i \in \mathbb{R}^{m \times 1}\}_{i=1}^l$, the symmetry matrices $\mathcal{S} = \{\frac{(\boldsymbol{u}_i \boldsymbol{u}_j^T + \boldsymbol{u}_j \boldsymbol{u}_i^T)}{2} \in \mathbb{R}^{m \times m}\}_{i=1,j=1}^{l,l}$ are also linear independent.*

The Proof is attached in the Appendix.

## 3 GENERAL BILINEAR PROJECTION

### 3.1 DECOMPOSITION OF MATRIX BASES

Given a set of rank-$k$ matrices $\mathcal{W}_k^{m \times n} = \{\mathbf{W}_p \in \mathbb{R}^{m \times n}\}_{p=1}^{mn}$, if those matrices are linear independent, then $\mathcal{W}_k^{m \times n}$ is a rank-$k$ complete base set of the matrix space $\mathbb{R}^{m \times n}$.

Theorem 1 demonstrates that the rank-1 matrix base set can be decomposed into two vector base sets. According to Theorem 1, we can derive the traditional bilinear projection $\mathbf{Y} = \mathbf{U}^T \mathbf{X} \mathbf{V}$ which is famous in the fields of linear algebra (Strang et al., 1993) and machine learning (Pirsiavash et al., 2009; Nie et al., 2018). The conclusion is presented in the following theorem.

**Theorem 2.** *If the coefficient of $\boldsymbol{X} \in \mathbb{R}^{m \times n}$ on the base $\boldsymbol{u}_p \boldsymbol{v}_q^T$ is calculated as $y_{pq} = Tr(\boldsymbol{X}^T \boldsymbol{u}_p \boldsymbol{v}_q^T)$, then the coefficients of $\boldsymbol{X}$ on the whole base set $\mathcal{W} = \{\boldsymbol{u}_p \boldsymbol{v}_q^T | \boldsymbol{u}_p \in \mathcal{U}, \boldsymbol{v}_q \in \mathcal{V}\}_{p=1,q=1}^{m,n}$ are $\{y_{pq}\}_{p=1,q=1}^{m,n}$ which can form a coefficient matrix $\boldsymbol{Y} \in \mathbb{R}^{m \times n}$ satisfying:*

$$\mathbf{Y} = \mathbf{U}^T \mathbf{X} \mathbf{V} \tag{6}$$

*where $\boldsymbol{U} = [\boldsymbol{u}_1, \cdots, \boldsymbol{u}_m] \in \mathbb{R}^{m \times m}$ and $\boldsymbol{V} = [\boldsymbol{v}_1, \cdots, \boldsymbol{v}_n] \in \mathbb{R}^{n \times n}$.*

The Proof is presented in the Appendix.

**Remark 1.** *Theorem 2 gives a way to decompose a matrix base set into two low-dimensional vector base sets. A base set corresponds to a linear projection in the linear space. Thus, the solution to find an appropriate projection in a matrix space (a high-dimensional space) can be transformed to find two vector projections in the low-dimensional vector spaces, which will alleviate the overfitting problem and save the computational sources, including running time and memory storage. Theorem 2 is a mathematical interpretation of why so many matrix-based algorithms modeled by the projection $\boldsymbol{Y} = \boldsymbol{U}^T \boldsymbol{X} \boldsymbol{V}$ (Pirsiavash et al., 2009; Nie et al., 2018; Fukui et al., 2016) work well. However, according to Theorem 2, the feature reduction algorithms based on Eq.(14) only find rank-1 matrix projections. Rank-1 projections are a small port of feasible projections. If the applications prefer high-rank projections, Eq.(14) will miss them and lead to a performance deficit.*

In the next section, we will explore a more general bilinear projection which is based on matrix bases with high rank, i.e., $k > 1$.

### 3.2 FORMULATION OF THE GENERAL BILINEAR PROJECTION

Consider the coefficient of an arbitrary matrix $\mathbf{X} \in \mathbb{R}^{m \times n}$ projected on a rank-$k$ matrix base $\mathbf{W}_p \in \mathcal{W}_k^{m \times n}$. The coefficient can be calculated as $y_p = Tr(\mathbf{X} \mathbf{W}_p^T) = Tr(\mathbf{U}_p^T \mathbf{X} \mathbf{V}_p)$ by decomposing $\mathbf{W}_p$ as $\mathbf{W}_p = \mathbf{U}_p \mathbf{V}_p^T$ where $\mathbf{U}_p \in \mathbb{R}^{m \times k}$ and $\mathbf{V}_p \in \mathbb{R}^{n \times k}$. Since $Tr(\mathbf{U}_p^T \mathbf{X} \mathbf{V}_p) = vec^T(\mathbf{X} \mathbf{V}_p) vec(\mathbf{U}_p) = vec^T(\mathbf{V}_p)(\mathbf{I}_k \otimes \mathbf{X}) vec(\mathbf{U}_p)$, the calculation of $y_p$ is equivalent to:

$$y_p = Tr\left((\mathbf{I}_k \otimes \mathbf{X}) vec(\mathbf{V}_p) vec^T(\mathbf{U}_p)\right) \tag{7}$$

where $\otimes$ represents the Kronecker product, $\mathbf{I}_k$ is the $k \times k$ identity matrix.

Eq.(7) indicates that the coefficient $y_p$ equals the coefficient of $\mathbf{I}_k \otimes \mathbf{X}$ projected on the matrix $vec(\mathbf{V}_p) vec^T(\mathbf{U}_p)$ in $\mathbb{R}^{mk \times nk}$. As the analysis presented in the previous section, the term

$vec(\mathbf{V}_p)vec^T(\mathbf{U}_p)$ consists of a set of free implicit variables that make the appropriate projecting directions hard to learn. To overcome this shortcoming, we employ the property presented in **Theorem 1** to separate the implicit variables from the linear independent projecting directions.

Since $mn \ll mnk^2$, the matrices $\{vec(\mathbf{V}_p)vec^T(\mathbf{U}_p)\}_{p=1}^{mn}$ are more likely located in a subspace of $\mathbb{R}^{mk \times nk}$ whose matrix base set can be decomposed into two vector base sets $\{\hat{\mathbf{u}}_i\}_{i=1}^{l_1}$ and $\{\hat{\mathbf{v}}_i\}_{i=1}^{l_2}$. Thus, for each $vec(\mathbf{V}_p)vec^T(\mathbf{U}_p)$, we can find a matrix $\mathbf{L}_p \in \mathbb{R}^{l_1 \times l_2}$ to hold the following equation:

$$vec(\mathbf{V}_p)vec^T(\mathbf{U}_p) = \sum_{i=1}^{l_2}\sum_{j=1}^{l_1}(\mathbf{L}_p)_{ij}\hat{\mathbf{v}}_i\hat{\mathbf{u}}_j^T \tag{8}$$

where $(\mathbf{L}_p)_{ij}$ is the $ij$-th element in the matrix $\mathbf{L}_p$, $l_1 \leq mk$ and $l_2 \leq nk$.

By substituting Eq.(8) in Eq.(7), we obtain a new equation to calculate $y_p$ by employing $\mathbf{L}_p$, $\{\hat{\mathbf{v}}_i\}_{i=1}^{l_2}$ and $\{\hat{\mathbf{u}}_j\}_{j=1}^{l_2}$. Assigning $\{y_p\}_{p=1}^h$ in a vector denoted by $\mathbf{y} = [y_1, \cdots, y_h]$, the **general bilinear projection** is presented as follows:

$$\mathbf{y} = \mathbf{P}^T vec(\mathbf{U}^T(\mathbf{I}_k \otimes \mathbf{X})\mathbf{V}) \tag{9}$$

where $k$ indicates the rank of the matrix base set, $\mathbf{X} \in \mathbb{R}^{m \times n}$, $\mathbf{U} = [\hat{\mathbf{u}}_1, \cdots, \hat{\mathbf{u}}_{l_1}] \in \mathbb{R}^{mk \times l_1}$, $\mathbf{V} = [\hat{\mathbf{v}}_1, \cdots, \hat{\mathbf{v}}_{l_2}] \in \mathbb{R}^{nk \times l_2}$, $\mathbf{P} = [vec(\mathbf{L}_1), \cdots, vec(\mathbf{L}_h)] \in \mathbb{R}^{(l_1 l_2) \times h}$. Worthy of note is that the sizes of $\mathbf{U}$ and $\mathbf{V}$ in Eq.(9) become $mk \times l_1$ and $nk \times l_2$ respectively, which are different from the ones presented before in this paper.

**Remark 2.** *Worthy of note is that $\boldsymbol{P}$ stores the free implicit variables mentioned in the previous section. In our proposed projection, they become explicit variables whose values are easy to constrain. Unlike Eq. (2), our bilinear projection considers all feasible projecting directions to reduce dimension and intra-class variance. Thus, our projection facilitates yielding more compact features than other FBiP approaches.*

**Remark 3.** *BiP methods MPN (Li et al., 2017a), MPN-COV(Wang et al., 2020), iSQRT-COV (Li et al., 2018), SMSO (Yu & Salzmann, 2018) and DBTNet-50 (Simonyan & Zisserman, 2014) reduce the dimension of local feature $\boldsymbol{x}_s$ by a linear projection $\hat{\boldsymbol{x}}_s = \boldsymbol{U}^T\boldsymbol{x}_s$ and then calculate low-dimension bilinear features as $\boldsymbol{U}^T\boldsymbol{x}_s\boldsymbol{x}_s^T\boldsymbol{U}$. According to our analysis, it equals to reduce the dimension of $\boldsymbol{x}_s\boldsymbol{x}_s^T$ by rank-$1$ matrix projecting directions, which misses much information. From Eq.(9), the projection on local features should be $\hat{\boldsymbol{x}}_s = \boldsymbol{U}^T(\boldsymbol{I}_k \otimes \boldsymbol{x}_s)$.*

The term $\mathbf{I}_k \otimes \mathbf{X}$ in Eq.(9) has a lot of zeros which cost lots of memory storage. By introducing the matrix partition as $\mathbf{U} = [\mathbf{U}_1^T, \cdots, \mathbf{U}_k^T]^T$ and $\mathbf{V} = [\mathbf{V}_1^T, \cdots, \mathbf{V}_k^T]^T$, we can reformulate our general bilinear projection as $\mathbf{y} = \mathbf{P}^T vec(\sum_{i=1}^k \mathbf{U}_i^T\mathbf{X}\mathbf{V}_i)$. where $\mathbf{U}_i \in \mathbb{R}^{m \times l_1}$, $\mathbf{V}_i \in \mathbb{R}^{n \times l_2}$ and $\mathbf{P} \in \mathbb{R}^{l_1 l_2 \times h}$, respectively.

# 4 RANK-$k$ FACTORIZED BILINEAR POOLING

## 4.1 FORMULATION

**Bilinear pooling.** Following traditional FBiP method (Kim et al., 2016), we employ the Eq.(9) to reduce the dimension of the bilinear feature $\mathbf{x}_s\mathbf{y}_t^T$, and formulate a new compact bilinear pooling method named rank-$k$ factorized bilinear pooling (RK-FBP) presented as follows.

$$\mathbf{f}_i = \mathbf{P}^T\left(vec\left(\sum_{r=1}^k(\mathbf{U}_r^T\mathbf{x}_s + \mathbf{b}_r^u)(\mathbf{V}_r^T\mathbf{y}_t + \mathbf{b}_r^v)^T\right)\right) + \mathbf{b}^p \tag{10}$$

where $\mathbf{b}_r^u \in \mathbb{R}^{l_1 \times 1}$, $\mathbf{b}_r^v \in \mathbb{R}^{l_2 \times 1}$ and $\mathbf{b}^p \in \mathbb{R}^{h \times 1}$ are the bias terms of projections $\mathbf{U}_r \in \mathbb{R}^{m \times l_1}$, $\mathbf{V}_r \in \mathbb{R}^{n \times l_2}$ and $\mathbf{P} \in \mathbb{R}^{h \times (l_1 l_2)}$, respectively.

**Remark 4.** *As seen from Eq.(10), let us set $l_1 = l_2 = h$. If we fix $\boldsymbol{P} \in \{0, 1\}^{h \times h^2}$ in which only the $(i, (i-1)h + i)$-th element equals "1". Then our model is equivalent to the bilinear pooling model used in (Gao et al., 2020; Amin et al., 2020). If we set the $(i, (i-1)h + i)$-th element of $\boldsymbol{P}$, i.e., $\boldsymbol{P}_{ik}$, as learnable parameters while other elements are set as "0", our model is equivalent to the bilinear pooling module used in (Kim et al., 2016; Lu et al., 2016; Yu et al., 2017).*

**Multi-linear pooling.** We employ the proposed bilinear projection to generate high-order pooling features. However, the dimension of high-order pooling features is huge, so it is intractable to

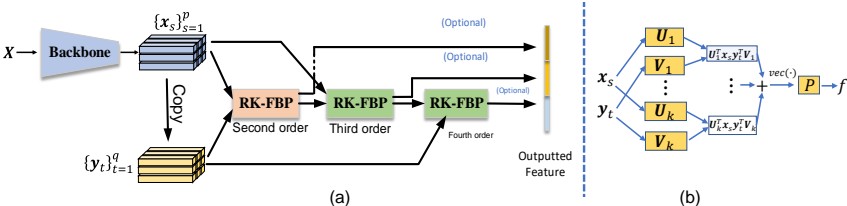

Figure 4: (a) The proposed multi-linear pooling structure. (b) RK-FBP module. In (a), we use three **Rk-FBP** modules to integrate high-order information into the multi-linear representations. From left to right, the 1-th, 2-th and last modules integrates the second-order information, the third-order information, and the fourth-order information, respectively. Each RK-FBP has its own parameters. At last, we choice several types of those features and concatenate them to a more informative one.

perform the dimension reduction on the high-order features directly. Thus, we give a recursive way to output the compact high-order pooling features by utilizing several RK-FBP modules. Without ambiguity, we denote $\mathbf{f}_i^t$ as the $t$-th order pooling feature, then the $(t + 1)$-th order pooling feature can be calculated as follows.

$$\mathbf{f}_i^{t+1} = \mathbf{P}^T \left( vec \left( \sum_{r=1}^{k} (\mathbf{U}_r^T \mathbf{x}_s + \mathbf{b}_r^u)(\mathbf{V}_r^T \mathbf{f}_i^t + \mathbf{b}_r^v)^T \right) \right) + \mathbf{b}^p \tag{11}$$

For obtained features $\{\mathbf{f}_i^t\}_{t=1}^O$ , we concatenate them into one feature vector, i.e., $\mathbf{f}_i = [(\mathbf{f}_i^1)^T, \cdots, (\mathbf{f}_i^O)^T]^T$ before feeding them into the classifier. Considering the information of different orders may be conflicted with each other, some order pooling features can be ignored in the concatenation. The structure of our model is depicted in Figure 4.

**Memory Analysis.** Compared with the fully bilinear pooling algorithm (Lin, 2015), our method results in a great saving of memory storage and computational load. For $c$-th classification tasks, **U**, **V** and **P** and the classifier hyperplane have $kml_1 + knl_2 + l_1l_2h + ch$ elements. In image classification tasks adopting ResNet50 as backbone, there are $m = 2048$, $n = 2048$, BiP requires $mnc = 200mn \approx 10^9$ parameters to output the classification result. In our RK-FBP, we set $l_1 = l_2 = 300$, $h = 512$ and $k = 4$ at most, there is about $10^7$ parameters, which is smaller than that of BiP by two orders of magnitude, and same as that of traditional factorized bilinear pooling algorithm, e.g., FBC (Gao et al., 2020; Fang et al., 2019), etc. Due to having a smaller amount of parameters, the computational load is also reduced. Because the high-order RK-FBP is recursively constructed by nesting several RK-FBP modules together, its storage memory is several times of the singular RK-FBP. Considering a $T$-fold multi-linear feature has $m^T$ dimension for a $m$-dimension first-order feature, the weights of our multi-linear version of RK-FBP are much smaller.

## 5 EXPERIMENTS AND ANALYSIS

### 5.1 EXPERIMENTAL SETTING

We conduct experiments on the image classification tasks to evaluate the proposed RK-FBP model. The adopted image datasets are Describing Texture Dataset (DTD) (Cimpoi et al., 2014), MINC-2500 (MINC) (Bell et al., 2015), MIT-Indoor (INDOOR) (Quattoni & Torralba, 2009), and Caltech-UCSD Bird (CUB200) (Xie et al., 2013), Cars196 (Krause et al., 2013), Aircraft (Maji et al., 2013) which are the texture image dataset in the wild, the material dataset, the indoor scene dataset, and two fine-grained image datasets, respectively. We incorporate our RK-FBP module in Deep Neural Networks as Figure 4, and implement it by PyTorchPaszke et al. (2017). ImageNet-pretrainings are taken from torchvision Marcel & Rodriguez (2010). The adopted backbones are VGG-16 and ResNet50. For those image data, each image is resized into $418 \times 418$. VGG-16 generates a $28 \times 28 \times 512$ feature map from $con5 - 3$ layer. The number of local features $d = 512$ and number of local feature is $784$. For ResNet50, it generates a $14 \times 14 \times 2048$ feature map which result $196$ 2048-dimensional features. Following Gao et al. (2016), the compact bilinear feature **f** are sent to the softmax classifier. Each RK-FBP module has its own parameters **U**, **V**, and **P**, which are updated by the back-propagation algorithm. The Pytorch is run on GPU Quadro RTX 6000. The optimization is done using SGD, weight decay is 0.01, learning rate is 0.01, momentum is 0.9. The number of epoch is 60 and batch size is 32.

## 5.2 ABLATION EXPERIMENT

We adopt VGG16 as the backbone for the ablation experiments.

**Rank $k$.** Different $k$ means different matrix bases, the compact bilinear features also have different performance. To demonstrate how the rank $k$ affects Rk-FBP, we perform RK-FBP on Indoor data set with varied $k \in [1, \cdots, 15]$. The number of column and row projections are set as $l_1 = l_2 = l \in \{50, 100, 150, 200, 250\}$ and the dimension of the bilinear feature is set to $h = 512$. The results are depicted in Figure. 6. We can find that While $l = 250$ and $k = 2$, RK-FBP achieves the best performance 83.5%. When $l = 250$ and $k = 1$, the accuracy is about 81.9% with about 2.6% decrease. When $l => 150$, the accuracy is relatively stable after $k > 2$. The accuracy variance of $k$ for each $l$ is within 2%. It means only $k = 2$ is enough. But when $l < 150$, the maximal accuracy corresponds large $k = 9$. This means when the number of matrix bases is less, a large rank is preferred. Because large rank matrices have more parameters, so they can depict the projecting directions more accurate. Thus, the observation may indicates fewer dimensions need more accurate projecting directions to capture the discriminant information. Besides, we employ

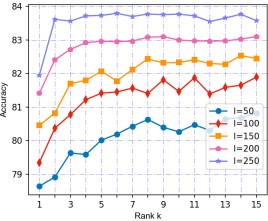

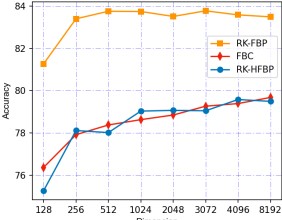

Figure 5: Accuracy of RK-FBP with different Rank $k$ on Indoor (dimension $h = 512$). When $l$ is small, $k$ should be large to achieve its largest accuracy.

Figure 6: Accuracy of RK-FBP with different dimensions on Indoor. FBC and RK-HFBP need large dimensions to achieve the best result.

$\hat{\mathbf{x}}_s = \mathbf{U}^T(\mathbf{I} \otimes \mathbf{x}_s)$ to improve the dimension procedure in SMSO (Li et al., 2017a), MPN(Yu & Salzmann, 2018), iSQRT-COV(Li et al., 2018). As seen from Table 1, we can find that increase of the rank $k$, the accuracy of those methods can be improved. And the maximum gain is 6.6% achieved by MPN on the Indoor dataset. Besides, we can find that with the increase of $k$, the accuracy will increase first and then be stable. This is consistent with our RK-FBP when the number of matrix bases is s small. Both experiments illustrate the importance of rank $k$. This indicates that current dimension reduction methods in those bilinear pooling models limit their expressive ability.

**Dimensionality $h$.** We compare our model with FBC, RK-FBP, and RK-HFBP (presented in Eq.(38)) on Indoor. We report the testing accuracy at $h = [128, 256, 512, 1024, 2048, 3072, 4096, 8192]$. We set the rank $k = 2$ for our RK-FBP and $l_1 = l_2 = 250$ for RK-FBP. For RK-HFBP and FBC, the rank of them are empirically set as $k = 8$ and $k = 14$, respectively. As seen from Figure 6, RK-FBP outperforms FBC and

Table 1: Accuracy (%) with different rank $k$ of the projection on different datasets.

|  | $k$ | 1 | 4 | 7 | 10 | 13 |
|---|---|---|---|---|---|---|
| SMSO | DTD | 69.3 | 71.3 | 70.9 | 71.5 | **71.7** |
|  | Indoor | 79.5 | 81.2 | 82.1 | **83.1** | 82.6 |
| MPN | DTD | 68.0 | 69.4 | 70.9 | **71.4** | 71.3 |
|  | Indoor | 76.5 | 78.9 | 79.2 | 82.6 | **83.1** |
| iSQRT-COV | DTD | 69.7 | 70.3 | 71.3 | **71.5** | 70.9 |
|  | Indoor | 79.5 | 80.3 | 81.9 | 82.3 | **83.4** |

RK-HFBP with different dimensionality. Specifically, RK-FBP is higher than FBC about 4.1% when $h = 512$. It validates that our model can find more discriminative projecting directions than RK-HFBP and FBC. After the dimension increase as $h > 512$, the performances of RK-FBP are relatively stable. Considering RK-FBP achieves the best result among the three approaches, this indicates that the performance of RK-FBP saturates. Nevertheless, for FBC and RK-HFBP, their performance continues to increase when $h > 512$. It may be because they can not find accurate projecting directions, so more discriminant information is acquired when the feature dimension increases. However, they can not surpass the RK-FBP. This implies that the missing discriminant information can not be completely solved by increasing the dimension of the compact bilinear features.

## 5.3 COMPARISON WITH THE STATE-OF-THE-ART ALGORITHMS

**VGG-16 backbone**. We first compare with full bilinear pooling methods: BCNN, improved BCNN, $DeepO2P$, $G^2DeNet$, RUN, DeepKSPD. Except BCNN, those methods adopt enhanced normalization strategies to overcome the bursitness problem. As seen from Table 2, only $G^2DeNet$ slightly surpasses our RK-FBP on CUB200 dataset by 0.6%. On the rest five datasets, our RK-FBP achieves the best results, and the largest gain is 8.6% (more than DeepO2P) on the MINC dataset.

Table 2: Comparisons for BiP methods in terms of Average Precision (%)

| Methods | Backbone | Feature dim. | Param | DTD | Indoor | MINC | CUB200 | Cars196 | Aircraft |
|---|---|---|---|---|---|---|---|---|---|
| VGG-16 (Simonyan & Zisserman, 2014) | VGG16 | 4096 | 120.4M | 60.1 | 64.5 | 73.0 | 80.4 | 76.7 | 74.1 |
| BCNN (Lin, 2015) | VGG16 | $131K$ | 52.4M | 67.5 | 77.6 | 74.5 | 84.0 | 91.1 | 87.1 |
| Improved BCNN(Lin & Maji, 2017) | VGG16 | 131K | 411M | - | - | - | 85.8 | 92.0 | 88.5 |
| DeepO2P (Ionescu et al., 2015) | VGG16 | $131K$ | 52.4M | 67.2 | 78.4 | 74.8 | - | - | - |
| CBP (Gao et al., 2016) | VGG16 | 8K | 1.64M | 67.7 | 76.8 | 73.3 | 84.0 | 90.1 | 87.1 |
| MPN (Li et al., 2017a) | VGG16 | 32K | 13.1M | 68.0 | 76.5 | 76.2 | 86.1 | 92.2 | 89.9 |
| $G^2$DeNet (Wang et al., 2017) | VGG16 | 131K | 411M | - | - | - | 87.1 | 92.5 | 89.0 |
| LRBP (Kong & Fowlkes, 2017) | VGG16 | - | 0.8M | 65.8 | - | - | 84.2 | 90.9 | 87.3 |
| FBN (Li et al., 2017b) | VGG16 | - | 0.39 | 67.8 | - | - | 82.9 | 87.7 | 84.8 |
| SMSO (Yu & Salzmann, 2018) | VGG16 | 2048 | 1.46M | 69.3 | 79.5 | 78.0 | 85.0 | 92.1 | 88.1 |
| RUN (Yu et al., 2020) | VGG16 | 131K | - | 68.4 | 80.8 | - | 86.3 | 81.0 | 89.8 |
| DeepKSPD (Engin et al., 2018) | VGG16 | 131K | - | - | 81.0 | - | 86.5 | 90.1 | 91.0 |
| ReDro (Rahman et al., 2020) | VGG16 | 33K | 103M | - | 80.2 | - | 86.7 | 92.2 | 91.0 |
| iSQRT-COV (Li et al., 2018) | VGG16 | 33K | 308M | 69.7 | 79.5 | 79.2 | 87.2 | 91.1 | 90.0 |
| HBP (Yu et al., 2018) | VGG16 | 24.6K | 39B | - | - | - | 87.2 | 93.7 | 90.3 |
| MoNet-2 (Gou et al., 2018) | VGG16 | 10K | 10M | - | - | - | 85.7 | 91.8 | 89.3 |
| FCBN (Yu et al., 2021) | VGG16 | 4K | - | 66.8 | - | - | 85.6 | - | 90.5 |
| FBC (Gao et al., 2020) | VGG16 | 8192 | 10M | 71.5 | 79.9 | 80.2 | 84.3 | 90.3 | 87.1 |
| TKBF (Yu et al., 2022) | VGG16 | $9K$ | 96 | 68.2 | 80.5 | 78.2 | 86.0 | 84.3 | 91.4 |
| RK-FBP ($2^+$, ours) | VGG16 | 512 | 45.2M | **72.2** | **83.5** | **83.4** | 86.8 | **92.5** | **91.9** |
| RK-FBP ($3^+ + 2^+$, ours) | VGG16 | 1024 | 52.3M | **74.1** | **85.1** | **84.3** | 87.4 | **93.6** | **92.5** |
| TKBF (Yu et al., 2022) | ResNet50 | 9K | 96 | 71.4 | 84.1 | 79.3 | 85.7 | 84.1 | 92.1 |
| DBTNet-50 (Zheng et al., 2019) | DBTNet | 2K | - | - | - | - | 87.5 | 94.5 | 91.2 |
| ReDro (Rahman et al., 2020) | ResNet50 | 33K | 103M | - | 80.2 | - | 86.7 | 92.2 | 91.0 |
| iSQRT-COV (Li et al., 2018) | ResNet50 | 33K | 312M | 70.4 | 80.1 | 81.3 | 87.9 | 92.1 | 90.9 |
| SMSO (Yu & Salzmann, 2018) | ResNet50 | 2K | 5.75M | 72.5 | 79.7 | 81.3 | 85.8 | 92.2 | 88.9 |
| RK-FBP ($2^+$, ours) | ResNet50 | 512 | 112M | **73.2** | **84.5** | **84.9** | 86.7 | **92.9** | **92.8** |
| RK-FBP ($3^+ + 2^+$, ours) | ResNet50 | 1024 | 112M | **74.9** | **86.2** | **85.1** | **88.1** | **94.5** | **93.6** |

Then, we further compare with the medium-scale bilinear features, ReDro and iSQRT-COV. ReDro is a grouped bilinear pooling method which reduces the dimension of bilinear features to 33K. As shown in Table 2, Our RK-FBP achieves comparable results with ReDR. Specifically, RK-FBP is more than ReDro by 0.1% and 0.3% CUB200 and Car196, and 3.3% and 0.9% on Indoor and Aircraft datasets, respectively. iSQRT-COV reduces the dimension of the local features from 512 to 256 using a $1 \times 1$ convolution layer. So, the dimension of its bilinear features is 33K too. Our RK-FBP outperforms iSQRT-COV on the rest five datasets except CUB200. Especially, RK-FBP significantly surpasses iSQRT-COV by 4.0% on Indoor. Because the ReDro and iSQRT-COV are designed to overcome the burstiness of bilinear features, their dimensions are relatively low.

We conclude two conclusions: (1) those positive results over the high dimensional BiP methods indicate that high dimension limits the efficiency of bilinear features in the succeeding classification tasks. (2) Considering that the above comparison methods employ matrix normalization strategies, the comparable results mean our RK-FBP can effectively solve the bursitness problem caused by the intra-class variances, that indicating that our method can find suitable projecting directions.

We also compare several state-of-the-art compact bilinear pooling methods: CBP, LRBP, FBC, and TKPF. As for those methods, we report the best results and their corresponding dimensions. As shown in Table 2, our RK-FBP achieves better classification accuracy using the lowest dimension. Compared to $4K$, the smallest dimensions of those comparison methods, 512 and 1024 are extremely compact. It is mainly because those compact methods can not find suitable projecting directions since they miss a lot of possible directions. As for the number of parameters, our RK-FBP uses more parameters than other compact methods. However, compared with the importance of accurate projecting directions, the cost of such an amount of storage is acceptable. Most of our model's parameters are contributed by **P**, whose parameters can be reduced by constraining it using sparsity regularization. It will be done in our future work.

At last, we report the performance of our multi-linear models ($3^+ + 2^+$ means capturing the 2-order and the 3-th order information). As seen from Table 2, our multi-linear model achieves the best accuracy while the dimension is 1024. Especially, our multi-linear features surpass HPB by 0.2%, where HBP is an enhanced bilinear pooling method fusing features across different layers by Hadamard product. Because the dimension of traditional compact bilinear pooling is high, it is hard to calculate multi-linear features by nesting them together. This excellent result shows the benefit of extremely low dimension features in the community.

**ResNet50 backbone.** As shown in Table 2, our models (bilinear and multi-linear) surpass other comparison methods on most datasets for the comparison methods, which indicates our model is robust to the backbones. Besides, we also compare with DBTNet-50 is a deep structure constructed by bilinear transformation blocks. We find that DBTNet-50 surpasses RK-FBP on CUB200 and Cars196 being weaker than RK-FBP on Aircraft. However, considering that DBTNet-50 applies the bilinear pooling in every layer, the results reported are not significant. Besides, our multi-linear features can outperform DBTNet-50 on most datasets with smaller dimensions. It may be because

the transformation used DBTNet-50 is based on the rank-1 matrix bases whose learning ability is limited. So DBTNet-50 may be improved by our proposed rank-$k$ bilinear projection.

## 6 CONCLUSION

In this paper, we reveal that traditional factorized bilinear pooling tends to miss feasible projecting direction. To overcome this disadvantage, we propose a general bilinear projection to formulate a pooling module called rank-$k$ factorized bilinear pooling (RK-FBP). Our RK-FBP has three advances: (1) RK-FBP is derived from a general bilinear projection based on complete matrix bases, so no feasible projecting directions will be missed. (2) Because the projecting directions are accurate, the learned bilinear features are not only compact but also discriminative. Those benefits give RK-FBP the power to produce more expressive compact bilinear features. Conducted experiments demonstrate that RK-FBP outperforms various state-of-the-art algorithms on challenging image classification tasks.

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

Appendix

# 7 DEFINITIONS AND PROOFS

**Definition 1:** *In the linear space $\mathbb{R}^{m \times n}$, the inner production between two matrices $A \in \mathbb{R}^{m \times n}$ and $B \in \mathbb{R}^{m \times n}$ is defined as $<A, B> = Tr(AB^T)$.*

**Definition 2:** *$\{W_p\}_{p=1}^{mn}$ is a group of complete bases of $\mathbb{R}^{m \times n}$, if and only if, for any matrix $X \in \mathbb{R}^{m \times n}$, there is an unique non-zero vector $y = [y_1, y_2, \cdots, y_{mn}]$ to hold the following equation:*

$$\mathbf{X} = \sum_{p=1}^{mn} y_p \mathbf{W}_p \tag{12}$$

*where $y_p$ is the projected coefficient of $X$ on the p-th base $W_p$.*

The vector $\mathbf{y}$ can be seen as the representation of the matrix $\mathbf{X}$ on the base set $\{\mathbf{W}_p\}_{p=1}^{mn}$.

**Definition 3:** *Suppose $\mathcal{W}_k^{m \times n} = \{W_p \in \mathbb{R}^{m \times n}\}_{p=1}^{mn}$ with $\max_p\{rank(W_p)\} = k$. $\mathcal{W}_k^{m \times n}$ is a complete orthonormal base set of the matrix space $\mathbb{R}^{m \times n}$, if the following equation holds :*

$$Tr(\mathbf{W}_p^T \mathbf{W}_q) = \begin{cases} 1 & p = q \\ 0 & p \neq q \end{cases} \tag{13}$$

**Theorem 1.** *Suppose $\mathcal{U} = \{u_p \in \mathbb{R}^{m \times 1}\}_{p=1}^{l_1}$ and $\mathcal{V} = \{v_q \in \mathbb{R}^{n \times 1}\}_{q=1}^{l_2}$ are two groups of linear independent vectors in two spaces $\mathbb{R}^{m \times 1}$ and $\mathbb{R}^{n \times 1}$, respectively. If the vector set $\mathcal{W} = \{vec(W_i) \in \mathbb{R}^{m \times n}\}_{i=1}^{mn}$ is constructed by $W_i = u_p v_q^T$ where $i = l_1(q-1) + l_2$, then $\mathcal{W}$ is a set of linear independent vectors.*

**Proof.** Suppose $\mathbf{U} = [\mathbf{u}_1, \cdots, \mathbf{u}_{l_1}]$ and $\mathbf{V} = [\mathbf{v}_1, \cdots, \mathbf{v}_{l_2}]$. Because $vec(\mathbf{W}_i) = vec(\mathbf{u}_p \mathbf{v}_q^T) = \mathbf{u}_p \otimes \mathbf{v}_q$, there is $\mathbf{U} \otimes \mathbf{V} = [vec(\mathbf{u}_1 \mathbf{v}_1^T), vec(\mathbf{u}_1 \mathbf{v}_2^T), \cdots, vec(\mathbf{u}_1 \mathbf{v}_{l_2}^T), \cdots, vec(\mathbf{u}_{l_1} \mathbf{v}_1^T), \cdots, vec(\mathbf{u}_{l_1} \mathbf{v}_{l_2}^T)]$. $\otimes$ is the Kronecker product. Thus, columns in $\mathbf{U} \otimes \mathbf{V}$ are the vectors in $\mathcal{W}$. According to the property of Kronecker product, there is $rank(\mathbf{U} \otimes \mathbf{V}) = l_1 l_2$. Thus, $\mathcal{W}$ is full rank, and the column in $\mathcal{W}$ is a group of linearly independent vectors.
□

**Theorem 2.** *If the coefficient of $X \in \mathbb{R}^{m \times n}$ on the base $u_p v_q^T$ is calculated as $y_{pq} = Tr(X^T u_p v_q^T)$, then the coefficients of $X$ on the whole base set $\mathcal{W} = \{u_p v_q^T | u_p \in \mathcal{U}, v_q \in \mathcal{V}\}_{p=1,q=1}^{m,n}$ are $\{y_{pq}\}_{p=1,q=1}^{m,n}$ which can form a coefficient matrix $Y \in \mathbb{R}^{m \times n}$ satisfying:*

$$\mathbf{Y} = \mathbf{U}^T \mathbf{X} \mathbf{V} \tag{14}$$

*where $U = [u_1, \cdots, u_m] \in \mathbb{R}^{m \times m}$ and $V = [v_1, \cdots, v_n] \in \mathbb{R}^{n \times n}$.*

**Proof.** Because $y_{pq} = Tr(\mathbf{X}^T \mathbf{u}_p \mathbf{v}_q^T) = \mathbf{u}_p^T \mathbf{X} \mathbf{v}_q$, so by arranging $Y_{pq}$ as a matrix, there is $\mathbf{Y} = \mathbf{U}^T \mathbf{X} \mathbf{V}$.
□

**Corollary.** *Given a set of linear independent vectors $\{\boldsymbol{u}_i \in \mathbb{R}^{m \times 1}\}_{i=1}^l$, the symmetry matrices $\mathcal{S} = \{\frac{(\boldsymbol{u}_i \boldsymbol{u}_j^T + \boldsymbol{u}_j \boldsymbol{u}_i^T)}{2} \in \mathbb{R}^{m \times m}\}_{i=1,j=1}^{l,l}$ are also linear independent.*

**Proof.** According to Theorem 2, we know the matrices $\{\mathbf{u}_i \mathbf{u}_j^T\}_{i=1,j=1}^{l,l}$ is a set of linear independent matrices. Thus, the solution of following function is $c_{ij} = 0$.

$$\sum_{i=1}^l \sum_{j=1}^l c_{ij} \mathbf{u}_i \mathbf{u}_j^T = \mathbf{0} \tag{15}$$

Let us consider the following equations with variables $\mathcal{C}' = \{c'_{ij} | i = 1, \cdots, l; j = 1, \cdots, l; i < j\}$.

$$\sum_{i=1}^l c'_{ii} \mathbf{u}_i \mathbf{u}_i^T + \sum_{i=1}^l \sum_{j<i} c'_{ij} \frac{(\mathbf{u}_i \mathbf{u}_j^T + \mathbf{u}_j \mathbf{u}_i^T)}{2} = \mathbf{0} \tag{16}$$

If $c'_{ij} = 0$, we let $c_{ij} = c'_{ij}/2$ for $i < j$ and $c_{ii} = c'_{ii}$. The function in Eq.(16) has an non-zeros solution. This violates the conclusion that $c_{ij} = 0$ for Eq.(16). Thus, all elements in $\mathcal{C}'$ are 0. It means the matrix set $\mathcal{S}$ is a group of linear independent matrices.

# 8 Discussion on Burstiness and Normalization Strategies

## 8.1 Why the Bilinear Pooling Can Enhance the Discriminant Ability of Local Features

Given a local feature $\mathbf{x}_i \in \mathbb{R}^{m \times 1}$, its corresponding bilinear feature is $\mathbf{z}_i = vec(\mathbf{x}_i \mathbf{x}_i^T)$. Let us calculate the inner product between bilinear features of $\mathbf{x}_i$ and $\mathbf{x}_j$, there is

$$< \mathbf{z}_i, \mathbf{z}_j >= vec(\mathbf{x}_i \mathbf{x}_i^T)^T vec(\mathbf{x}_j \mathbf{x}_j^T) = (\mathbf{x}_i^T \mathbf{x}_j)^2 \tag{17}$$

Comparing Eq.(17) with the polynomial kernel function $k(\mathbf{x}_i, \mathbf{x}_j) = (a(\mathbf{x}_i^T \mathbf{x}_j) + d)^p$, the inner product between bilinear features equals to polynomial kernel function with $a = 1$, $d = 0$ and $p = 2$. Therefore, we can claim that the bilinear feature is the explicit result of the non-linear projection determined by the polynomial kernel function.

How to let the features outputted by the backbone fit well to the hyper-parameters of polynomial kernel function is not the concern of our paper. Thus, we suppose the backbone can generate features good enough to satisfy the hyper-parameters.

As for the local features $\mathbf{X}_i = [\mathbf{x}_{i1}, \mathbf{x}_{i2}, \cdots, \mathbf{x}_{ic}] \times \mathbb{R}^{m \times c}$, the bilinear pooling $\mathbf{Z}_i = \mathbf{X}_i \mathbf{X}_i^T \in \mathbb{R}^{m \times m}$ is just the sum of bilinear features of columns in $\mathbf{X}_i$. So its enhanced discriminant ability can be also interpreted by the polynomial kernel function.

As well known, the polynomial kernel function can improve the classification performance of the support vector machines which is a linear discriminant classifier. Thus, the discriminant information in bilinear features can be well depicted by the linear projection.

This is the basis that we can employ linear projections to reduce the dimensionality of bilinear features, which is a crucial step to solve the burstiness problem of bilinear features.

## 8.2 How the Burstiness Reduce the Performance of models

Burstiness phenomenon on the image features is first analysed in the literature Jégou et al. (2009), which focus on the feature vectors obtained by the bag-of-words frameworks.

In bag-of-words frameworks, each dimension of feature vectors corresponds to an visual word collected from the whole image data. For the $i$-th image, the feature vector is $\mathbf{x}_i = [x_{i1}, x_{i2}, \cdots, x_{im}]^T \in \mathbb{R}^{m \times 1}$ where the value of $x_{ij}$ is the frequency of the $j$-th visual word appeared in the $i$-th image. Suppose the label of $i$-th image is $y_i$, so the average value of the $j$-th

dimension of the whole $c$-th class of samples is $\bar{x}_j = \frac{1}{N_c} \sum_{y_i=c} x_{ij}$ where $N_c$ is the sample number of the $c$-th class.

In some cases, the $j$-the visual word may appear a lot of times in the $i$-th image but does not appear so many times in other images. This will make the value of $x_{ij}$ much larger than the average value of the $j$-th dimension, i.e., $\bar{x}_j$. This is the burstiness phenomenon of bag-of-words features. Geometrically, this burstiness phenomenon increases the variance of samples in each class in the feature space.

Therefore, literature Wei et al. (2018) describes the burstiness phenomenon as "*the problem that the feature descriptor is not invariant enough where the feature elements may have large variances within the same class.*"

Thus, the burstiness problem can be summarized as the problem of large intra-class variance.

The burstiness of bilinear features is caused by the outer product on the local features. Let us take the three-dimensional data as an example. Suppose there are three samples $\mathbf{x}_1 = [1,1,1]^T$, $\mathbf{x}_2 = [3,1,1]$, $\mathbf{x}_3 = [1.2, 1.2, 1.2]^T$. The bilinear features of $\mathbf{x}_1$, $\mathbf{x}_2$, and $\mathbf{x}_3$ are $\mathbf{z}_1 = vec(\mathbf{x}_1\mathbf{x}_1^T) = [1,1,1,1,1,1,1,1,1]^T$, $\mathbf{z}_2 = vec(\mathbf{x}_2\mathbf{x}_2^T) = [9,3,3,3,1,1,3,1,1]^T$, and $\mathbf{z}_3 = vec(\mathbf{x}_3\mathbf{x}_3^T) = [1.44, 1.44, 1.44, 1.44, 1.44, 1.44, 1.44, 1.44, 1.44]^T$.

We can calculate the average of $\mathbf{z}_1$ and $\mathbf{z}_3$, i.e., $\bar{\mathbf{z}} = \frac{\mathbf{z}_1 + \mathbf{z}_2}{2} = [1.22, 1.22, 1.22, 1.22, 1.22, 1.22, 1.22, 1.22, 1.22]^T$. If most of local features are close to $\mathbf{x}_1$ and $\mathbf{x}_3$, $\bar{\mathbf{z}}$ can be considered as the average bilinear feature of the whole class. Thus, $\mathbf{z}_2$ is far away from the average $\mathbf{z}$. Because this phenomenon is similar to the burstiness of bag-of-words features, it is also called as the burstiness of bilinear features, which also expands the intra-class variance.

For some images, there are the illumination variations and appearance changes in them Gao et al. (2020); Wei et al. (2018), which make the features extracted by deep neural networks also have some variances within each class. The variance may reflect the singular values of the matrix storing local features, i.e., $\mathbf{X} = \sum_{i=1}^{T} \mathbf{u}_i \sigma_i \mathbf{v}_i^T$ where $\sigma_i$ is the $i$-the singular value and $\mathbf{u}_i$ and $\mathbf{v}_i$ are the corresponding singular vectors. Because the bilinear pooling on $\mathbf{X}$ is $\mathbf{X}\mathbf{X}^T = \sum_{i=1}^{T} \mathbf{u}_i \sigma_i^2 \mathbf{u}_i^T$. Those variances will be expanded by the outer product and cause the burstiness of bilinear features in the deep frameworks.

Because the burstiness will affect the similarity between bilinear features Wei et al. (2018), it will affect the performance of models based on similarity.

For classification tasks, the bilinear features are likely linear discriminant. Due to the high dimensionality of bilinear features, there are a lot of feasible solutions of classifiers can fit the training data well. However, the large intra-class variance caused by the burstiness may let the classifier select a bad solution which has the bad generalization on the test dataset, and the performance of bilinear features is harmed. Thus, how to alleviate the burstiness problem is very important for learning the bilinear features.

### 8.3 SIGNED ELEMENTWISE SQUARE-ROOT OPERATION

Signed elementwise square-root operation transforms a vector $\mathbf{x} = [x_1, x_2, \cdot, x_m]^T$ to a new vector $\hat{\mathbf{x}} = [\hat{x_1}, \hat{x_2}, \cdots, \hat{x_m}]^T$, in which $\hat{x}_i$ is calculated as

$$\hat{x}_i = sgn(x_i)\sqrt{|x_i|} \tag{18}$$

Consider the value of $\sqrt{|x_i|}$, there is

$$\begin{cases} \sqrt{|x_i|} >= |x_i|, & |x_i| <= 1 \\ \sqrt{|x_i|} < |x_i|, & |x_i| > 1 \end{cases} \tag{19}$$

In this way, we can find that $\sqrt{|x_i|}$ let the value $|x_i|$ close to 1. Let us consider the value $sgn(x_i)$, Eq.(18) lets the vector $\mathbf{x} \in \mathbb{R}^{m \times 1}$ close to the centers $[\pm 1, \pm 1, \cdots, \pm 1] \in \mathbb{R}^{m \times 1}$ where the symbol $\pm$ is determined by $sgn(x_i)$.

Thus, if samples from different classes are well separated and are located in different quadrants in the feature space, the elementwise square-root operation can reduce the intra-class variance well.

And the generalization is good. Such a requirement can be satisfied by the bag-of-words features. It is because the bag-of-words frameworks generate feature vectors according the frequency of each visual elements appeared in each image, where the visual elements are often generated by clustering algorithms and thus have explicit similarity meanings.

The literature Wei et al. (2018) reveals the features extracted by the deep neural networks do not meet the requirement signed Elementwise Square-root transformation.

### 8.4 $L_2$-NORMALIZATION

$L_2$-normalization is widely used in the deep neural networks to enhance the generalization ability of learned features. For a vector $\mathbf{x}_i \in \mathbb{R}^{m \times 1}$, the $L_2$-normalization of $\mathbf{x}_i$ is $\hat{\mathbf{x}}_i$ defined as

$$\hat{\mathbf{x}}_i = \mathbf{x}_i / |\mathbf{x}_i|_2 \tag{20}$$

Therefore, the Euclidean distance between two $\hat{\mathbf{x}}_i$ and $\hat{\mathbf{x}}_j$ is $|\hat{\mathbf{x}}_i - \hat{\mathbf{x}}_j|_2^2 = 2 - \hat{\mathbf{x}}_i^T \hat{\mathbf{x}}_j$. Thus, after $L_2$-normalization, the Euclidean distance between samples can be replaced by the cosine distance. For the features extracted by deep neural networks, $L_2$-normalization will reduce the variances in each class Meng et al. (2021), because the variance information is along the radius direction in the feature space.

### 8.5 HOW THE FACTORIZED BILINEAR POOLING ALLEVIATE THE BURSTINESS

Because the bilinear features are linear discriminant, so the dimension of bilinear features can be reduced by a set of linear projections. In this way, the variance along those eliminated directions are discarded. Then, the $L_2$-normalization strategy is adopted to reduce the variance along the radius direction in the low-dimensional feature space. In this way, the intra-class variances are reduced and the burstiness problem is alleviated.

## 9 RELATED WORK

We review works improving bilinear pooling in two aspects: 1) Enhancing the bilinear feature's effectiveness; 2) Reducing the dimension of the bilinear feature for greater efficiency.

**Effectiveness improvement.** There are several different routines to improve the bilinear feature's effectiveness. For example, G2DeNet (Wang et al., 2017), FASON (Dai et al., 2017) and MoNet (Gou et al., 2018) take the first-order statistics into consideration beside the bilinear pooling. KP (Cui et al., 2017), HOK (Koniusz et al., 2016) and HOP (Cherian et al., 2017) extend the second-order pooling to higher-order pooling. By considering bilinear pooling as depicting correlations across different features in a specific kernel space, Ker-RP (Wang et al., 2015; Zhang et al., 2021) employs kernel functions instead of inner-product to strengthen the representative capability of the bilinear feature. Bilinear pooling can be enhanced by capturing correlations between features yielded by different layers (Yu et al., 2018). Besides, the strategies of normalization are also proved to be effective in improving the discriminant ability of bilinear features. For instance, improved bilinear pooling (IBP) (Lin & Maji, 2017), MPN-COV (Li et al., 2018) and their variants (Koniusz et al., 2018) explore the matrix normalization to moderate the singular values of the bilinear matrix. By using those strategies, the discriminant ability of bilinear features is increased by a large margin compared with the original bilinear features.

**Dimension reduction.** The bilinear feature is a high dimensional matrix, which limits its efficiency and makes it prone to over-fitting. To speed up the classification and suppress over-fitting, FBN (Li et al., 2017b) reduces the dimension of the weight matrices in the classifier by replacing each weight matrix with a product of two low-rank matrices. In parallel, some methods focus on the dimension reduction for the original first-order features before bilinear pooling. BCNN (Lin, 2015) uses principal component analysis (PCA) to learn the projection matrix for reducing the dimension of first-order features. Specifically, it uses the projection matrix learned from PCA as initialization of a $1 \times 1$ convolution layer and then trains the network in an end-to-end manner. The PCA is also utilized in LRBP (Kong & Fowlkes, 2017) and iSQRT-COV (Li et al., 2018) for dimension reduction on the first-order features. Except those linear projection with learnable parameters, CBP

(Gao et al., 2016; Yu et al., 2021) approximates the operation of bilinear pooling by the polynomial kernel function. Then CBP employs two kernel approximation methods to reduce the dimension of bilinear feature, tensor sketch and random Maclaurin (RM), which achieve better performance than the PCA. Inspired by CBP, the strategy of CBP is adopted in MLB (Fukui et al., 2016) for cross-modal understanding. In CBP, the projections is formulated to reduce the dimension of the first-order features by two random projections which is then fused by Hadamard product. Hadamard product-based low-rank factorized bilinear pooling (HFBP) (Kim et al., 2016) replaces the random matrices with two learnable matrices and obtains a new compact bilinear pooling algorithm. Such technique is then adopted by HBP (Yu et al., 2018) for fusing features in different layers. Although so many dimension reduction algorithms have been proposed in past years, there lacks a general perspective to understand them. In this paper, we review the dimension reduction algorithms from the perspective of finding appropriate projection directions. Moreover, we reveal that the projections used in those algorithms tend to miss a lot of possible projecting directions, which reduces the effectiveness of the compact bilinear feature.

## 9.1 INTERPRETATION OF BILINEAR FEATURES FROM THE PERSPECTIVE OF SUPERVISED SPECTRAL GRAPH PARTITIONING

### 9.1.1 SPECTRAL GRAPH PARTITIONING

Let us consider the similarity between two samples $\mathbf{x}$ and $\mathbf{y}$ with polynomial kernel function $K(\mathbf{x}, \mathbf{y}) = (\gamma(\mathbf{x}^T \mathbf{y}) + d)^2$ where $\gamma$ and $d$ are two hyper-parameters adjusted for different data distributions. If we calculate the inner product between two bilinear features $vec(\mathbf{x}\mathbf{x}^T)$ and $vec(\mathbf{y}^T\mathbf{y}^T)$: $< (\mathbf{x}\mathbf{x}^T, \mathbf{y}\mathbf{y}^T >= (\mathbf{x}\mathbf{y}^T)^2$, we can find that it is the special case of the similarity calculated by the polynomial kernel function with $\gamma = 1$ and $d = 0$. The similarity matrix $\mathbf{S}$ on a data set can be used to describe the discriminant information between

As well known, the polynomial kernel function can be adopted to solve the inseparable problem by support vector machines (SVMs) and $k$-means algorithms. The ability to solve the inseparable problem also enhance the discriminant ability of vectors $\mathbf{x}$.

Let us adopt the spectral graph partitioning as a tool to analyze the performance of bilinear features. Before doing this, we introduce the procedure of spectral graph partitioning.

**Definition 4.** Given a set of samples $\{\mathbf{x}_i\}_{i=1}^N$, the adjacent matrix $\mathbf{S}$ is defined by

$$\mathbf{S}_{ij} = \begin{cases} K(\mathbf{x}_i, \mathbf{x}_j), & i \neq j \\ 0, & i = j \end{cases} \tag{21}$$

where $K(\mathbf{x}_i, \mathbf{x}_j)$ is a kernel function whose value monotonically decreases with respective to the distance between $\mathbf{x}$ and $\mathbf{y}$, e.g., $K(\mathbf{x}, \mathbf{y}) = exp(-\gamma|\mathbf{x} - \mathbf{y}|_2^2)$ $(\gamma > 0)$.

The normalized Laplace matrix is defined as $\mathbf{L} = \mathbf{I} - \mathbf{D}^{-1/2}\mathbf{S}\mathbf{D}^{-1/2}$ where $\mathbf{D}$ is a diagonal matrix with $\mathbf{D}_{ii} = \sum_{j=1} \mathbf{S}_{ij}$. The following equation is held:

$$\mathbf{f}^T \mathbf{L} \mathbf{f} = \sum_{i=1}^N \sum_{j=1}^N \mathbf{S}_{ij} \left(\frac{f_i}{\sqrt{d_i}} - \frac{f_j}{\sqrt{d_j}}\right)^2 \tag{22}$$

where $\mathbf{f} = [f_1, f_2, \cdots, f_N]^T \in \mathbb{R}^{N \times 1}$. For traditional bi-class spectral graph partitioning, the objective function is to minimize Normalized cuts whose the objective function is presented as follows.

$$\min_{\mathbf{f} \in \{-1, 1\}^{N \times 1}} \sum_{i=1}^N \sum_{j=1}^N \mathbf{S}_{ij} |\frac{f_i}{\sqrt{d_i}} - \frac{f_j}{\sqrt{d_j}}|_2^2$$

Because the distance between $\mathbf{x}_i$ and $\mathbf{x}_j$ is smaller, the value of $k(\mathbf{x}_i, \mathbf{x}_j)$ is larger, the solution in the above optimization problem is

$$f_i = \begin{cases} \sqrt{d_i}, & x_i \in \mathcal{C}_1 \\ -\sqrt{d_i}, & x_i \in \mathcal{C}_2 \end{cases} \tag{23}$$

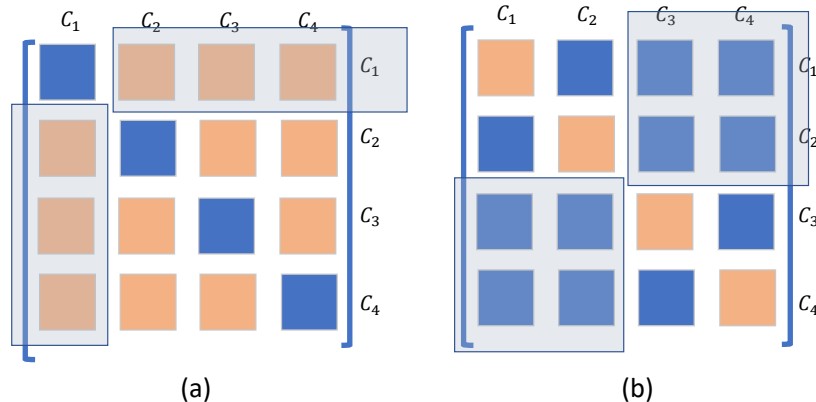

Figure 7: (a) Traditional spectral graph partitioning. (b) Our designed spectral graph partitioning. There are four classes $\{C_i\}_{i=1}^4$. When we use the bi-class based model to deal with the fourth classes, the partitioning of two models are different. In (a), the first class consists of $C_1$, and the second class consists of $\{C_2, C_3, C_4\}$; in (b), the first class consists of $\{C_1, C_2\}$, and the second class consists of $\{C_3, C_4\}$. When we extend the bi-class model to multi-class version, the strategy is different.

where $\mathcal{C}_i$ is the collection of samples in the $i$-th class.

The above is the traditional graph partitioning model. In our graph partitioning model, we let the kernel function $k(\mathbf{x}_i, \mathbf{x}_j)$ whose value monotonically increases with respective to the distance between $\mathbf{x}_i$ and $\mathbf{y}_j$. Thus, the objective function of our spectral clustering can be formulated as follows.

$$\max_{\mathbf{f} \in \{-1,1\}^{N \times 1}} \sum_{i=1}^N \sum_{j=1}^N \mathbf{S}_{ij} |\frac{f_i}{\sqrt{d_i}} - \frac{f_j}{\sqrt{d_j}}|_2^2 \tag{24}$$

Because the task only has two classes, we can obtain the solution presented in Eq.(23) by maximizing the objective function.

However, the two graph partitioning algorithms presented in Eq.(23) and Eq.(24) behavior quite different when they are employed to solve the multi-class tasks. To clearly show the difference, we suppose there are 4-classes of samples. The $i$-th class is denoted as $C_i$, and the samples from the same class are listed together. The kernel matrix $\mathbf{S}$ of two methods are graphically shown in Figure 7. Both methods consider the $\mathbf{f}^T \mathbf{L} \mathbf{f}$ which equals sum of values in the dark-colored boxes in Figure 7. Because the tradition model minimizes the value of $\mathbf{f}^T \mathbf{L} \mathbf{f}$, so it tends to find dark-colored boxes consisting smaller areas. As for our model, it maximizes $\mathbf{f}^T \mathbf{L} \mathbf{f}$, so it finds the dark-colored boxes having the largest area. In this way, the two classes obtained by the traditional graph partitioning model are $\{C_1\}$ and $\{C_2, C_3, C_4\}$, which indicates that it separates one class of samples from the reset classes. When we want to separate the 4 classes, we need 4 vectors $\{\mathbf{f}_i\}_{i=1}^4$ to indicates the class-membership of samples.

However, as for our model, the obtained classes consist of samples in $\{C_1, C_2\}$ and $\{C_3, C_4\}$, which is not consistent to the ground-truth of samples.

But, if we further separate $C_1$ from $C_2$, and $C_3$ from $C_4$, i.e., obtain another partition $\{C_1, C_3\}$ and $\{C_2, C_4\}$, we can partition the data set well. Such a partitioning result corresponds to $f_1 = [\sqrt{d_1}, \sqrt{d_2}, -\sqrt{d_3}, -\sqrt{d_4}]$ and $f_2 = [\sqrt{d_1}, -\sqrt{d_2}, \sqrt{d_3}, -\sqrt{d_4}]$. Obviously, the two dimensional samples $[\sqrt{d_1}, \sqrt{d_1}], [\sqrt{d_2}, -\sqrt{d_2}], [-\sqrt{d_3}, \sqrt{d_3}], [-\sqrt{d_4}, \sqrt{d_4}]$ are the centers of the four classes, and they are separated well. Thus, we can employ those two vectors $\{\mathbf{f}_i\}_{i=1}^2$ to extend the bi-class model to a 4-class model.

And so on, for a $C$-class task, our model only need to employ $k = log_2(C)$ vectors $\{\mathbf{f}_i\}_{i=1}^k$ to embedding the original samples. Compared with the traditional graph partitioning model needing $C$ vectors, the embedding of our model is extremely compact.

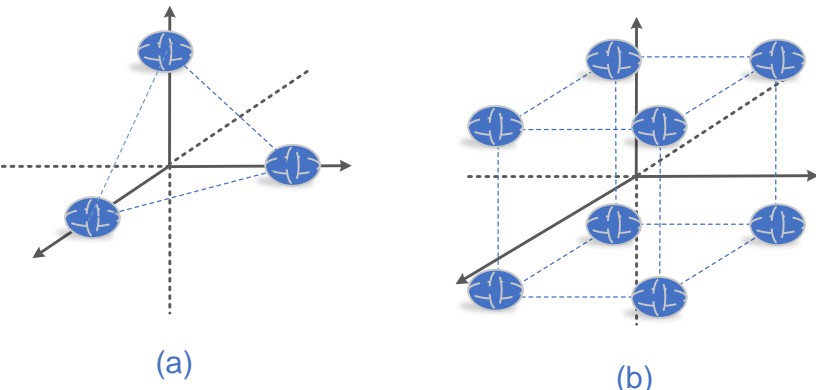

Figure 8: (a) Traditional spectral graph partitioning. (b) Our designed spectral graph partitioning. The traditional graph partition method needs $C$ dimension to code $C$ classes of samples, while our method can only use $log_2(C)$ dimension to code $C$ classes of samples.

**Remark 5.** *Our proposed graph partitioning algorithm can only employ $log_2(C)$ dimensions to represent the embeddings of samples. Theoretically, we can employ 64-dimension to code $2^{64}$ classes of samples. The geometrical illustration of samples obtained by two types of graph partitioning methods are shown in the Figure 8.*

Now, we employ our proposed graph partitioning algorithm to extract the features. Our mult-class graph partitioning model is formulated as

$$
\max_{\mathbf{F}^T\mathbf{F}=\mathbf{D}_k} \sum_{k=1}^{C}\sum_{i=1}^{N}\sum_{j=1}^{N} \mathbf{S}_{ij}|\frac{f_i^k}{\sqrt{d_i}} - \frac{f_j^k}{\sqrt{d_j}}|_2^2
$$
$$
\iff \max_{\mathbf{F}^T\mathbf{F}=\mathbf{D}_k} Tr(\mathbf{F}^T(\mathbf{I} - \mathbf{D}^{-\frac{1}{2}}\mathbf{S}\mathbf{D}^{-\frac{1}{2}})\mathbf{F})
$$
$$
\iff \max_{\mathbf{F}^T\mathbf{F}=\mathbf{D}_k} Tr(\mathbf{F}^T(-\mathbf{D}^{-1/2}\mathbf{S}\mathbf{D}^{-1/2})\mathbf{F}) \tag{25}
$$
$$
\iff \max_{\mathbf{F}^T\mathbf{F}=\mathbf{D}_k} \|(\mathbf{D}^{-1/2}\mathbf{S}\mathbf{D}^{-1/2}) - \mathbf{F}\mathbf{F}^T\|_F^2
$$

where $k = log_2(C)$.

**Remark 5**. The solution of $\mathbf{F}$ in Eq.(25) is the $log_2(C)$-th smallest eign-vectors of $\mathbf{D}^{-1/2}\mathbf{S}\mathbf{D}^{-1/2}$.

**Lemma.** If the sample $N$ is large enough, the eigen-vectors of $\mathbf{D}^{-1/2}\mathbf{S}\mathbf{D}^{-1/2}$ are equal to the eigen-vectors of $\mathbf{S}$. Thus, the embedding $\mathbf{F}$ can be obtained by performing eign-value decomposition on $\mathbf{S}$.
**Proof:** Because $ii$-th element in $\mathbf{D}^{-1/2}$ is $d_i = \frac{1}{\sqrt{\sum_{j=1}^{N}\mathbf{S}_{ij}}}$. Because, the $\mathbf{S}_{ij}$ is the value of $k(\mathbf{x}_i, \mathbf{x}_j)$ which is monotonically increase with distance between $\mathbf{x}_i$ and $\mathbf{x}_j$. If $N$ is large enough, the error $\|d_i - d_j\| < \epsilon$. Under this assumption, there is that $d_1 \approx d_2 \approx d_3 \approx \cdots \approx d_N$. Therefore, we can let $\mathbf{D}^{-1/2}\mathbf{S}\mathbf{D}^{-1/2} \approx (d_i)^2\mathbf{S}$. Thus, we have that the embedding $\mathbf{F}$ can be approximated by the find the $log_2(C)$-th smallest eign-vectors $\mathbf{S}$.

**Theorem 3.** *When $X$ is the bilinear features, $S = X^TX$ becomes the similarity matrix determined by the second-order polynomial kernel function. Let us denote $F$ as the solution in Eq.(25), thus there exists a linear projection $L$ to let $F = L^TX$.*

**Proof.** Because $\mathbf{S} = \sum_{i=1}^{r}\sigma_i\mathbf{u}_i\mathbf{u}_i^T$, where $\sigma_i$ is the $i$-th small eigen-value of $\mathbf{S}$, thus, $\sum_{i=1}^{k}\sigma_i\mathbf{u}_i\mathbf{u}_i^T = \mathbf{F}\mathbf{F}^T$. Because $\mathbf{S} = \mathbf{X}^T\mathbf{X}$, there is $\mathbf{X} = \sum_{i=1}^{r}\sqrt{\sigma_i}\mathbf{u}_i\mathbf{v}_i^T$, there is $\sum_{i=1}^{k}\sqrt{\sigma_i}\mathbf{v}_i^T = [\mathbf{u}_1, \mathbf{u}_2, \cdots, \mathbf{u}_k]^T\mathbf{X} = \mathbf{F}$.

**Remark.** The above theorem prove that the bilinear features can be reduced to an extremely low dimensional space with there discriminant information being preserved. This is the theoretical base why our method can reduce the bilinear features to 512.

## 10    RELATION WITH EXISTING METHODS

### 10.1    PCA BASELINE

PCA baseline is exploited in BCNN and iSQRT-COV for dimension reduction for bilinear features. It adopts a $1 \times 1$ convolutional layer to reduce the dimension of the local feature from $m$ to $l_1$ ($l_1 < m$). Then the bilinear pooling is performed on the $l_1$ dimensional features to $l_1^2$-dimensional feature. To be specific, given $N$ local features $\hat{\mathbf{X}} = [\mathbf{x}_1, \mathbf{x}_2, \cdots, \mathbf{x}_N] \in \mathbb{R}^{m \times N}$, PCA baseline generate the compact local features by $\mathbf{Y} = \mathbf{U}^T \hat{\mathbf{X}}$. Thus, the compact bilinear feature is obtained by

$$\mathbf{Z} = \mathbf{U}^T \hat{\mathbf{X}} \hat{\mathbf{X}} \mathbf{U} \tag{26}$$

According to the theorem 1, the Eq.(26) actually employs a set of rank 1 matrix bases to reduce the high-dimensional bilinear feature $\hat{\mathbf{X}}\hat{\mathbf{X}}$ to a compact one $\mathbf{Z}$. According to our analysis, if the data prefers large rank matrix bases, some discriminant information is lost in the dimension reduction procedure.

To overcome this shortcoming, we should employ $k$ projecting matrices $\{\mathbf{U}_t\}_{t=1}^k$ to reduce the local feature $\hat{\mathbf{X}}$ and calculate the compact bilinear feature $\mathbf{Z}$ as follows:

$$\mathbf{Z} = \sum_{t=1}^{k} \mathbf{U}_t^T \hat{\mathbf{X}} \hat{\mathbf{X}} \mathbf{U}_t \tag{27}$$

### 10.2    RANDOM MACLAURIN (RM)

RM is employed in CBP (Gao et al., 2016) for compact bilinear pooling. inspired by the success of CBP, many RM-based algorithms are proposed (Yu et al., 2021; Fukui et al., 2016). RM employed two random projecting matrices $\mathbf{U}$ and $\mathbf{V}$ to reduce the dimension of bilinear features.

$$\mathbf{z} = \sum_{i=1}^{N} \mathbf{U}^T \mathbf{x}_i \circ \mathbf{U}^T \mathbf{x}_i \tag{28}$$

where $\mathbf{z}$ is the compact bilinear feature, and $\circ$ is the Hadamard product.

Let us compare Eq.(28) with Eq.(4). If we set $\mathbf{P}$ as an identity matrix, Eq.(28) with Eq.(4) are the same. The difference is that $\mathbf{U}$ and $\mathbf{V}$ in Eq.(4) are updated by gradient descent algorithms while those in Eq.(28) are random variables found by sampling values from the random distributions. It may be why methods of Eq.(4) outperform methods based on RM in terms of classification accuracy. However, because the parameters in Eq.(4) do not need to update via gradient descent algorithm, it involves less computation.

### 10.3    TWO-LEVEL KRONECKER-PRODUCT PRODUCT FACTORIZATION (TKPF)

TKPF supposes a projecting matrix $\mathbf{P}$ can be decomposed as the Kronecker product presented as follows.

$$\mathbf{P} = \sum_{q=1}^{Q} \mathbf{A}^{(q)} \otimes \mathbf{B}^{(q)} \tag{29}$$

Then, by further decomposing $\mathbf{A}^{(q)}$ and $\mathbf{B}^{(q)}$ as $\mathbf{A}^q = \mathbf{I}_r \otimes \hat{\mathbf{A}}^{(q)}$ and $\mathbf{B}^{(q)} = \mathbf{I}_r \otimes \hat{\mathbf{B}}^{(q)}$, TKPF formulates the compact bilinear pooling as

$$\mathbf{Z} = \sum_{q=1}^{Q} (\mathbf{I}_r \otimes (\hat{\mathbf{B}}^{(q)})^T) \mathbf{X} \mathbf{X}^T (\mathbf{I}_r \otimes \hat{\mathbf{A}}^{(q)}) \tag{30}$$

where $\mathbf{A}^{\hat{(q)}}$ and $\mathbf{B}^{\hat{(q)}}$ are learnable parameters. Because the scale of $\mathbf{A}^{\hat{(q)}}$ and $\mathbf{B}^{(q)}$ can be adjusted by the parameter $r$, TKPF can use very less parameters to reduce the dimension of bilinear feature.

However, we can find the TKPF has the following shortcoming.

TKPF assumes that every matrix $\mathbf{P}$ can be decomposed as Eq.(29). However, this assumption is not true. If we construct the $i$-th column of $\mathbf{P}$ as $\mathbf{p}_i = \sum_{r=1}^{k} \mathbf{u}_i^r \otimes \mathbf{v}_i^r$, where $\mathbf{U}_r = [\mathbf{u}_1^r, \cdots, \mathbf{u}_d^r]$, $\mathbf{V}_r = [\mathbf{v}_1^r, \cdots, \mathbf{v}_d^r]$. Consider $\mathbf{U}^r \otimes \mathbf{V}^r$ equals

$$
\begin{aligned}
\mathbf{U}_r \otimes \mathbf{V}_r &= [\mathbf{u}_1^r \otimes \mathbf{V}_r, \mathbf{u}_2^r \otimes \mathbf{V}_r, \cdots, \mathbf{u}_d^r \otimes \mathbf{V}_r] \\
&= [\mathbf{u}_1^r \otimes \mathbf{v}_1^r, \mathbf{u}_1^r \otimes \mathbf{v}_2^r, \cdots, \mathbf{u}_d^r \otimes \mathbf{v}_d^r]
\end{aligned}
\tag{31}
$$

$$
\mathbf{P} = \sum_{r=1}^{k} [\mathbf{u}_1^r \otimes \mathbf{v}_1^r, \mathbf{u}_2^r \otimes \mathbf{v}_2^r, \cdots, \mathbf{u}_d^r \otimes \mathbf{v}_d^r]
\tag{32}
$$

Compared Eq.(31) with Eq.(32), we know $\mathbf{P} = [\mathbf{p}_1, \cdots, \mathbf{p}_d]$ can not be decomposed as Eq.(29). Besides, $\mathbf{A}^q = \mathbf{I}_r \otimes \hat{\mathbf{A}}^{(q)}$ is also a Kronecker product-based decomposition, which may be not hold. This means KTPF ignores a lot of feasible projecting directions. If the data prefers those missed feasible projecting directions, the performance is compact is bad.

Let us compare TKPF with our bilinear model. For easy comparison, we transform our bilinear projection $\mathbf{y} = \mathbf{P}^T vec(\sum_{i=1}^{k} \mathbf{U}_i^T \mathbf{X} \mathbf{V}_i)$ into a vector-based form presented as follows.

$$
\mathbf{y} = \mathbf{L}^T (\sum_{q=1}^{Q} \mathbf{V}_q^T \otimes \mathbf{U}_q^T) vec(\mathbf{X}\mathbf{X}^T)
\tag{33}
$$

Then, the projecting matrix in Eq.(33) is

$$
\hat{\mathbf{P}} = \mathbf{L}^T (\sum_{q=1}^{Q} \mathbf{V}_q^T \otimes \mathbf{U}_q^T)
\tag{34}
$$

Obviously, Eq.(29) and Eq.(34) look similar. The difference between them is that Eq.(33) has a matrix $\mathbf{L}$ while Eq.(29) does not. $\mathbf{L}$ plays the role to select the columns in $\mathbf{V}_q^T \otimes \mathbf{U}_q^T$ to form a matrix can not be decomposed by Kronecker product. This makes our $\hat{\mathbf{P}}$ can be any matrix. Therefore, the minimal difference makes our proposed projection is an accurate one while the projection in Eq.(29) is not.

Our proposed bilinear projection is general, it can be used to analyze the performance of other dimension reduction algorithms for matrix data, such as two-dimensional principal analysis (Zhang & Ren, 2011) and two-dimensional linear discriminant analysis (Ye et al., 2004).

As discussed above, TKPF employs some inappropriate matrix decompositions to construct the projection, which means TKPF also can not find accurate projecting matrices. Although the performance of TKPF looks good on its reported datasets, TKPF may suffer from a great performance reduction in other applications. Thus, the application range of TKPF is limited. At last, compared with our proposed bilinear model, the dimension of the compact bilinear feature is still high, e.g., for its best accuracy, the dimension is $96 * 96$.

### 10.4  CODE PERSPECTIVE OF BILINEAR POOLING

FBC Gao et al. (2020) proposes a general compact bilinear model from the coding perspective. Given $h$ low rank atoms $\{\mathbf{V}_i \mathbf{U}_i^T\}$, FBC Gao et al. (2020) encodes $\mathbf{x}_s \mathbf{y}_t^T$ into $\mathbf{f}_i \in \mathbb{R}^{h \times 1}$ by solving the following matrix-based sparse coding optimization problem:

$$
\min_{\mathbf{f}} \|\mathbf{x}_s \mathbf{y}_t^T - \sum_{l=1}^{h} f_l \mathbf{U}_l \mathbf{V}_l^T\|_F^2 + \lambda \|\mathbf{f}\|_1
\tag{35}
$$

where $\lambda$ is a trade-off between the reconstruction error and the sparsity. $\mathbf{U}_l \in \mathbb{R}^{m \times k}$ and $\mathbf{V}_l \in \mathbb{R}^{n \times k}$ are two rank-$k$ matrices decomposed from the $l$-th rank-$k$ matrix atom. Here, $\mathbf{U}_l$ and $\mathbf{V}_l$ are learned by the deep model through the whole set of original samples. The optimization problem in Eq.(35) has a closed-form solution presented as follows:

$$
\begin{cases}
\mathbf{f}' = \mathbf{P}((\mathbf{U}^T \mathbf{U} \mathbf{P}^T \circ \mathbf{V}^T \mathbf{V} \mathbf{P}^T))^{-1} \mathbf{P}(\mathbf{U}^T \mathbf{x}_s \circ \mathbf{V}^T \mathbf{y}_t) \\
\mathbf{f} = sign(\mathbf{f}') \circ \max(abs(\mathbf{f}') - \frac{\lambda}{2}, 0)
\end{cases}
\tag{36}
$$

where $\mathbf{U} = [\mathbf{U}_1, \cdots, \mathbf{U}_h] \in \mathbb{R}^{m \times hk}$ and $\mathbf{V} = [\mathbf{V}_1, \cdots, \mathbf{V}_h] \in \mathbb{R}^{n \times hk}$ are the learnable parameters of the dictionary. $\mathbf{P} \in \mathbb{R}^{h \times hk}$ is a fixed binary matrix with only elements in the row $l$, columns $((l-1) \times h) + 1$ to $(lh)$ being "1", where $l \in [1, h]$.

Because the above formulation involves the matrix inverse operation, it adopts a relaxation strategy $((\mathbf{U}^T \mathbf{U} \mathbf{P}^T \circ \mathbf{V}^T \mathbf{V} \mathbf{P}^T))^{-1} \mathbf{P}(\mathbf{U}^T \mathbf{x}_s \circ \mathbf{V}^T \mathbf{y}_t) = (\hat{\mathbf{U}}^T \mathbf{x}_s \circ \hat{\mathbf{V}}^T \mathbf{y}_t)$.

$$\begin{cases} \mathbf{f}' = \mathbf{P}(\hat{\mathbf{U}}^T \mathbf{x}_s \circ \hat{\mathbf{V}}^T \mathbf{y}_t) \\ \mathbf{f} = sign(\mathbf{f}'_i) \circ \max(abs(\mathbf{f}'_i) - \frac{\lambda}{2}, 0) \end{cases} \tag{37}$$

However, we can prove that the coding model is mathematically equivalent to the traditional model. And we can propose that a new coding-based model does not need relaxation. We consider the vector-based coding model $\min_\mathbf{y} \|\mathbf{x} - \mathbf{W}^T y\|_F^2$ where $\mathbf{y}$ is the coefficient on of vector $\mathbf{x}$ on the dictionary $\mathbf{W}$. There is a solution $\mathbf{y} = (\mathbf{W}\mathbf{W}^T)^{-1}\mathbf{W}\mathbf{x}$. According to FBC, when we replace the $\mathbf{x}$ as $vec(\mathbf{x}\mathbf{y}_t^T)$, the $i$-th column of $\mathbf{W}$ as $\mathbf{V}_i \mathbf{U}_i^T$, and $\mathbf{y}$ as $\mathbf{f}$, we can obtain the formulation Eq.(35).

Similarly, we can construct another coding-model $\min_\mathbf{y} \|\mathbf{x} - \mathbf{W}^T(\mathbf{W}\mathbf{W}^T)^{-1}\mathbf{y}\|_F^2$. $\mathbf{y}$ is the coefficient of $\mathbf{x}$ on the atoms $\mathbf{W}(\mathbf{W}^T\mathbf{W})^{-1}$. There is the solution $\mathbf{y} = \mathbf{W}^T\mathbf{x}$ which is a linear projection. Thus, according to the Eq.(36), if we replace $\mathbf{W}'$ the $i$-th column $\mathbf{w}_i$ as $\mathbf{V}_i \mathbf{U}_i^T$, the matrix-based dictionary is $\mathbf{P}((\mathbf{U}^T \mathbf{U} \mathbf{P}^T \circ \mathbf{V}^T \mathbf{V} \mathbf{P}^T))^{-1}\mathbf{P}\mathbf{R}^T$. The $i$-th column of $\mathbf{R}$ is $\mathbf{u}_i \otimes \mathbf{v}_i$ where $\mathbf{u}_i$ and $\mathbf{v}_i$ are the $i$-th column of $\mathbf{U}$ and $\mathbf{V}$. This is because $\mathbf{R}^T vec(\mathbf{x}_s \mathbf{y}_t^T) = \mathbf{U}^T \mathbf{x}_s \circ \mathbf{V}^T \mathbf{y}_t$. Because $\mathbf{u}_i \otimes \mathbf{v}_i = vec(\mathbf{u}_i \mathbf{v}_i^T)$, so $\mathbf{P}((\mathbf{U}^T \mathbf{U} \mathbf{P}^T \circ \mathbf{V}^T \mathbf{V} \mathbf{P}^T))^{-1}\mathbf{P}\mathbf{R}^T$ can also be transformed to a set of matrices. Thus, we have another type of matrix-based coding model, and the bilinear feature can be outputted by $\mathbf{f} = \mathbf{P}^T(\mathbf{U}^T \mathbf{x}_s \circ \mathbf{V}^T \mathbf{y}_t)$ which is equivalent to the formulation Eq.(37). And the solution is an accurate one.

This is why we directly derive the bilinear pooling model from our general bilinear projection $\mathbf{f} = \mathbf{P}^T vec(\mathbf{U}^T \mathbf{x}_s \mathbf{y}_t^T \mathbf{V})$ other than the coding based framework.

## 10.5 Formulation of RK-HFBP

Eq.(4) has a matrix $\mathbf{P} \in \mathbb{R}^{l \times h}$, so $l$ can not be very large. Thus, Eq.(4) is a rank-1 Hadamard product-based FBiP. As discussed in FBC, the rank of projecting matrices are also very important. So we improve Eq.(4) by giving its projecting matrices more rank. By employing the same strategy of our proposed rank-$k$ bilinear projection $Z = \mathbf{U}^T(\mathbf{I}_k \otimes \mathbf{X})\mathbf{V} = \sum_{i=1}^{k} \mathbf{U}_i^T \mathbf{X} \mathbf{V}_i$, we obtain a new rank-$k$ Hadamard product-based bilinear pooling (RK-HFBiP) presented as follows.

$$\mathbf{y} = \mathbf{P}^T vec(\sum_{i=1}^{k} \mathbf{U}_i^T x_s \circ \mathbf{V}_i y_t^T) \tag{38}$$

We will RK-HFBiP as a comparison method in our ablation experiments.

## 11 Ablation Experiments

Table 3: Accuracy (%) with different orders on variance datasets. $(a, \cdots, c)$ means the feature is constructed by concatenating the $a$-th,$\cdots$, $c$-th order statistic information.

| $T$ | 2 | 3 | 4 | (2,3) | (2,4) | (3,4) | (2,3,4) |
|---|---|---|---|---|---|---|---|
| Indoor | 83.8 | 83.1 | 82.1 | 85.1 | 85.2 | 84.9 | 85.0 |
| MNIC | 83.4 | 83.5 | 83.0 | 84.3 | 84.5 | 83.7 | 84.2 |
| CUB-200 | 86.8 | 86.9 | 86.7 | 87.4 | 87.2 | 87.2 | 87.6 |
| Cars-196 | 92.5 | 92.1 | 92.6 | 93.8 | 94.0 | 93.6 | 93.9 |

**Number of orders**. We can nest several RK-FBP modules together to learn compact representations with high-order information, which we denoted as RK-FBP-M. We evaluate the influence of the order $T$ on RK-FBP-M. We set $l_1 = l_2 = h = 512$, and vary $d$ among $\{2, 3, 4\}$. Because our model allows us to concatenate the features with different orders, we have 7 types of features. As seen from Table 3, the multi-linear features win the bilinear features by a significant margin. Besides,

with the increase of the order, the result is not always increased. To be specific, the $(2,3)$, $(2,4)$, and $(2,3,4)$ features achieve the similar accuracy. Thus, in our paper, we set the order parameter as $(2,3)$ for our the multi-linear model.

**Normalization Strategy.** In the fully bilinear pooling approaches, the performance is crucially dependent on the normalization strategy. Thus, we explore how the normalization strategies affect our proposed RK-FBP. Thus, we add two normalization strategies adopted by fully bilinear pooling Lin (2015) and improved bilinear pooling Lin & Maji (2017) before RK-FBP modules. We denote them by 'SgnSqrt' normalization and 'SgnSqrt+log' normalization, respectively. We also compare the result with only 'log' normalization. **At last, all bilinear features should be normalized to an 'unit' vector by $L_2$ normalization**. The results are shown in the Table 4. As seen from the results, we can find that the results are similar with little variance. This is a bit different from Table 1, which employs normalization strategies after projection. In Table 1 the combination of feature reduction and normalization strategies increases the performance. This may be because the features after projection is still high, there are still much information of large intra-class variance. So the normalization strategies can improve the performance of models by removing the bad information. Because our RK-FBP does not use 'SgnSqrt' and 'log' but achieves comparable results with those methods using normalization strategies, it indicates our proposed model can solve the burstiness problem caused by large intra-class variances.

Table 4: Accuracy (%) with normalization strategies on variance datasets.

|  | Indoor | MNIC | CUB-200 | Cars-196 |
| --- | --- | --- | --- | --- |
| RK-FBP | 83.8 | 83.4 | 86.8 | 92.5 |
| SgnSqrt | 83.7 | 83.5 | 86.9 | 92.3 |
| log | 83.9 | 83.7 | 86.6 | 92.7 |
| SgnSqrt+log | 84.0 | 83.5 | 86.7 | 92.6 |

