# OpenReview forum: "Compact Bilinear Pooling via General Bilinear Projection"
_ICLR.cc/2023/Conference — Submitted to ICLR 2023_

### Official Review · Reviewer_MKth · 2022-10-22

**Confidence:** 5
**Correctness:** 3
**Technical Novelty And Significance:** 3
**Empirical Novelty And Significance:** 2
**Recommendation:** 6

**Clarity, Quality, Novelty And Reproducibility:**

The paper is generally very written. The idea on parallel 1x1 convolutions for reducing feature dimension before bilinear pooling may be a feasible solution. The proposed method is easy to implement. More evaluation and discussion seem be necessary.

**Strength And Weaknesses:**

Strength:

+: The idea on parallel 1x1 convolutions for reducing feature dimension before bilinear pooling seems bring clear improvement (Fig.5 and Table 1), which may be a feasible solution to balance performance and model complexity for bilinear pooling methods.

+: On several small-scale fine-grained image benchmarks, the proposed method achieves competitive results with a relatively low representation dimension.

+: The proposed method is clearly written and is easy to implement.

Weaknesses:

-: Could $\mathbf{U}$ and $\mathbf{V}$ in Eq. (10) be shared?

-: The idea on combination of 1x1 convolutions and fully-connected layer for obtaining compact covariance representations appeared in [r1] (Section 3.5 and Table 4), where sequential 1x1 convolutions (not parallel) are used to reduce feature dimension before covariance pooling, and then a fully-connected layer is adopted after covariance pooling. It would better discuss relationship between the proposed method and [r1].

[r1] Deep CNNs Meet Global Covariance Pooling: Better Representation and Generalization. IEEE T-PAMI, 2021.

-: The concerns on experiments.

(1)	Are hyper-parameters $k$ and $h$ sensitive to various datasets? Particularly, could the proposed method be flexibly adopted to large-species classification (e.g., iNat2017 [r2]) with the same hyper-parameters?

[r2] The iNaturalist species classification and detection dataset. CVPR, 2018.

(2)	I wonder what meaning of ‘Param’ in Table 2. Particularly, why Param of iSQRT-COV with backbone of ResNet-50 is 312M? To our best knowledge, iSQRT-COV is parameter-free itself. For dimension reduction, a 1x1 convolution (2048, 256) is used and contain about 0.5M parameters. If ‘Param’ in Table 2 contains one of classifier, it clearly varies for different datasets, while RK-FBP (2+) and RK-FBP (2+,3+) should have different parameters.

(3)	Besides parameters, FLOPs and running time are important metric to evaluate model complexity.

(4)	I would like to know how about performance of the proposed method under training-from-scratch setting (e.g., ImageNet-1K).


**Summary Of The Paper:**

This work aims to study how to effectively achieve compact bilinear representations. To this end, the authors proposed a low-rank factorized bilinear pooling method. Specifically, as shown in Eq. (10), $k$ parallel 1x1 convolutions are used to reduce feature dimension before bilinear pooling, and then $k$ bilinear representations are summed following by a fully-connected layer. As such, size of bilinear representations could be controlled by 1x1 convolution and fully-connected layer. The experiments are conducted on several small-scale fine-grained image benchmarks.

**Summary Of The Review:**

I have reviewed this work in previous conference, and the current version achieves clear improvement on writing and experiments. However, discussion on previous works and experimental evaluation could be further strengthened. Particularly, the authors should clarify meaning of ‘Param’ in Table 2.

---

> ### Author Response · Authors · 2022-11-19
> **Response to reviewer MktH(2)**
>
> ### Q5: What is the meaning of ‘Param’.
> ***Ans***: 'Param' means the number of parameters after the backbone. It includes the parameters of RK-FBP and the parameters of classifier. In the paper, we set the class number $200$. We will clarify it in the revised paper.
> Because we give the code for pruning the parameters of $y=P^Tvec(U^T(I_k\otimes X)V)$, the parameters of the proposed method can be further reduced. For example, we can let $P$ be the sparse matrix in which only $6%$ elements are non-zeros. In this way, the number of parameters and FLOPs will be reduced in the testing stage. For fair comparison, we do not report the result in the paper.
> ### Q6: The FLOPs and running time are also important for evaluating the algorithms.
> ***Ans***: The FLOPs is related to the number of float computations. We compare with float computation of our proposed bilinear projection with the Hadamard product-based projection.
>
> Given a set of local features $\{x_i\in \mathbb{R}^{m\times 1}\}$. For each local feature $x_i$, our projectiong is $y = P^Tvec((U^T(I_k\otimes x_i)+b_u)(b_v+(I_k\otimes x_i^T)V))+ b_p$ where $U\in \mathbb{R}^{mk\times l_1}$, $V \in \mathbb{R}^{mk\times l_2}$ and $P\in \mathbb{R}^{(l1l2)\times h}$.
>
> For the matrix multiplication of a $m\times n$ matrix and a $n\times d$ matrix, there are $mnd$ times multiplications and $m(n-1)d$ additions.
>
> Therefore, the FLOPs of the proposed method is $k((l_1+l_2)(2m)+l_1l_2)+l_1l_2h*2$ in which we treat the multiplication and addition equal.
>
> The Hadamard product bilinear projection is $y = P^T((U^Tx+b_u) \circ  (V^Tx + b_v)) + b_p$ where $U\in \mathbb{R}^{m\times kl} $,$V\in \mathbb{R}^{n\times kl} $ and $P\in \mathbb{R}^{kl\times h_1}$. Thus, the FLOPs of Hadamard product is $kl(4m+h_1+1)$.
> If we set $l_1=l_2=l$ and $h = h_1$, the FlOPs is much smaller than the proposed rank-$k$  bilinear projection. This is because, the $P$ in rank-$k$ bilinear projection is much larger than the $P$ in the Hadamard Product-based bilinear projection. In reality, the $l_1=l_2$ should much smaller than $l$ and $h$ is smaller than $h_1$, so the runing time of the rank-$K$ bilinear projection is similar to that of Hadamard product-based bilinear projection.
>
> We provide the running times of the compared methods as follows. The time is recorded by minute.
> |Backbone|Methods|CUB|	Cars|	indoor|
> |:----|:----|:----|:----|:----|
> |Resnet|     RK-FBP|	174.1|	319.1|	224.4|
> |Resnet|	        FBC|	170.3|	336.3|	310.6|
> |VGG	|RK-FBP|	149.4|	286.5|	217.6|
> |VGG	|FBC        | 152.0	|307.5	|232.1
>
> ### Q7: I would like to know how about the performance of the proposed method under training-from-scratch setting (e.g., imageNet-K)
> ***Ans***: We try to train our model on the imageNet-1K. But, due to the lack of computing resources, the training is very slow. One epoch needs about one day in our server. Despite we could not train the model on Imagenet-1K, we train the model on CUB-200-2021 from scratch. The result is $83.3$ which is a bit less than the result of using pre-trained model ($86.7$ is reported in our paper with ResNet50).

---

### Official Review · Reviewer_xdH1 · 2022-10-24

**Confidence:** 4
**Correctness:** 2
**Technical Novelty And Significance:** 2
**Empirical Novelty And Significance:** 2
**Recommendation:** 3

**Clarity, Quality, Novelty And Reproducibility:**

Clarity and quality of the paper should be improved. Section 2 and 3 are hard to follow. The presentation should be organized in a more structured way. The novelty is incremental as it seems like an extension of low-rank bilinear pooling (Kong & Fowlkes, 2017). It is unclear how to reproduce the results as the paper does not discuss how to address the burstiness of bilinear features and does not provide open-source code.



**Details Of Ethics Concerns:**

No ethics issues as I am aware.

**Strength And Weaknesses:**

Strength
- Improving bilinear features is important as standard bilinear features (via outer product) is very high-dimensional.
- Datasets used in the experiments are good.
- The way to produce compact multi-linear features is interesting.

Below are weaknesses.

- The paper does not discuss how to address the burstiness of bilinear/multi-linear features. To note, for bilinear features and low-rank bilinear features (Lin, 2015; Kong & Fowlkes, 2017), it is crucial to apply signed square root transform and L2 normalization.

- It is confusing to state "(RK-FBP) does not miss any projecting directions" in the abstract. Authors should clarify, e.g., how to define "any directions".

- In Introduction, it is unclear how existing factorized bilinear pooling (FBiP) methods address the burstiness problem. It seems the paper only mentions that these methods aim to reduce dimensionality of bilinear features. Can authors discuss?

- Figure 1 is confusing. In this example, is it just a Fisher Discriminant Analysis? There is not much insight from this figure.

- Figure 2: For methods that use Hadamard product, they learn U and V to optimize classification accuracy, which is the goal of the classification problem. Then, what does it mean by "missing values" in Hadamard project.

- Section 4: The proposed Rank-K Factorized Bilinear Pooling (RK-FBP) seems like an extension of low-rank bilinear pooling (Kong & Fowlkes, 2017). This makes the novelty incremental.


**Summary Of The Paper:**

The paper focuses on bilinear pooling in fine-grained image classification tasks. Bilinear pooling (that generates second-order features) is an effective method to capture details of fine-grained classes and achieves better performance than first-order features. Because bilinear features are high in dimension, many bilinear pooling methods adopt Hadamard product-based bilinear projection to reduce dimensionality. However, the paper argues that this misses "a lot of possible projecting directions which will significantly harm the performance". The paper then proposes low-rank factorized bilinear pooling (RK-FBP). The paper argues that RK-FBP "does not miss any projecting directions". The paper further extends RK-FBP to pooling features of multiple layers and forms multi-linear features. Experiments show the proposed methods achieves the state-of-the-art.

**Summary Of The Review:**

The paper improves over low-rank bilinear pooling (Kong & Fowlkes, 2017) so the novelty is limited. It does not discuss how to address burstiness of bilinear features, particularly multi-linear features. The literature reports that normalizing bilinear features (e.g., using signed square root and L2 normalization) is crucial. Presentation of the paper should be improved further. Therefore, the paper is rated as Reject.

---

> ### Author Response · Authors · 2022-11-13
> **Response to of Reviewer xdH1 (1)**
>
> ### Q1: The paper does not discuss how to address the burstiness of bilinear/multi-linear featrures. To note, for bilinear features and low-rank bilinear features, it is crucial to apply signed-square-root transform and $L_2$ normalization.
> **A** We did the discussion in the introduction of the original paper. We stated that the burstiness problem is caused by the large intra-class variance and can be solved by the dimension reduction. In the appendix section 10, we also compared our model with different types of normalization strategies to solve the burstiness problem. The experiment results are presented in the Table 4. In the experimental settings, our model also adopted the $L_2$ normalization.
>
> ### Q2:  Can authors discuss how FBiPs address the burstiness problem which makes the signed square-root transformation and L_2 normalization ( we use "RT + $L_2$"  for short) crucial?
>
> **A**: FBiPs employs the combination of linear dimension reduction and $L_2$ normalization to solve the burstiness problem.
>
> In literature [r2,r3], burstiness is described as the large intra-class variance, which will reduce the generalization ability of the bilinear features. It is because a smaller intra-class variance in local features will lead to a large intra-class variance in the bilinear features, thus the prediction is not stable.
>
> Thus, the main idea to deal with the burstiness is to reduce the intra-class variance of bilinear features.
>
> The procedure has two steps:
> - The linear projection selects appropriate projecting directions to remove the dimensions that do not help the discriminant analysis. This way, the intra-class variance kept in those eliminated dimensions is removed. This can be achieved because bilinear features are linear discriminant. (the bilinear pooling equals the non-linear projection determined by the polynomial kernel function $k(x,y)=(<xy^T>)^2$ [r1], the bilinear features are likely linear discriminant. This is the reason why bilinear features outperform the local features.)
>
> - After the dimension reduction, the samples of different classes are distributed around the origin point. Thus, the $L_2$-normalization can reduce the intra-class variance kept in the remaining dimensions.
>
>  The graphical illustration is shown in Figure 1 in the revised manuscript. As seen from Figure 1, the intra-class variance of the FbiP is much smaller than the intra-class variance of the combination of signed squared root and $L_2$ normalization. This indicates that our proposed model can solve the burstiness problem well without signed squared root transformation.
>
> The combination of signed squared root transformation and $L_2$ normalization is not good because the signed-squared-root transformation is designed for bag-of-word features, which is not suitable for the features extracted by deep neural networks [r1,r2].
>
> For the solution of the burstiness problem, we give a more thorough discussion in the appendix of the revised manuscript.
> - [r1] Compacted Bilinear Pooling. CVPR2016
> - [r2] On the Burstiness of Visual Elements. CVPR2009
> - [r3] Grassmann pooling as compact homogeneous bilinear pooling for fine-grained visual classification. ECCV2018.
>
> ### Q3: “RK-FBP does not miss any projecting directions” is confusing.
> **A**: We revise it as “RK-FBP considers the feasible projecting directions missed by Hadamard product-based bilinear pooling.”
>
> ### Q4: There is not much insight in Figure 1.
> **A**: We present a new figure 1 consisting of 6 sub-figures to compare the FBiP and the "RT + $L_2$" strategy for solving the burstiness problem.
>
> ### Q5: In figure 2, what does it mean by “missing values” in U and V connected by Hadamard product?
> **A**: Suppose $U = [u_1,u_2, \cdots, u_h]\in \mathbb{R}^{d\times h}$ and $V= [v_1,v_2,\cdots, v_h] \in \mathbb{R}^{d\times h}$. The projecting directions in the Hadamard product-based projection are $\{vec(u_1v_1^T), vec(u_2v_2^T) ,\cdots, vec(u_hv_h^T)\}$. Because the columns in $U$ and $V$ learned by practical optimization algorithms are not duplicated (see figure 3), i.e, $u_i \neq u_j$ (for $i \neq j$), $v_i \neq v_j$ ( for $i \neq j$), the directions $u_iv_i^T$, $u_iv_j^T$ and $u_iv_k^T$ will be not learned at the same time. We prove that $u_iv_i^T$, $u_iv_j^T$ and $u_iv_k^T (i\neq j\neq k)$ are linear independent, thus, they may be preferred by the training data for projecting. Because optimization algorithms can not learn them, they are missing values.

---

### Official Review · Reviewer_yyVh · 2022-10-25

**Confidence:** 4
**Correctness:** 3
**Technical Novelty And Significance:** 3
**Empirical Novelty And Significance:** 3
**Recommendation:** 6

**Clarity, Quality, Novelty And Reproducibility:**

Clarity
--------
This paper should improve readability.

P3 above Eq.(4):  “replace 1 in Eq.(2) by a learnable vector $p_r$”, but Eq.(4) is $P$ and $_r$ is not used for $U$ and $V$.

P5, 6 In the general bilinear projection in Eq.(9), the dimensions of $P, V, U$ differ from the previous sections. To clarify this fact, it would be better to use different notations.

Sec.10
Normalization: The discussion of 2DLDA seems to be irrelevant to the results of the normalization strategy.

There are typos, eg.,

P3 Theorem 1. $R^{m\times n}$ seems to be $R^{mn \times 1}$

P7 Rk-FBP

P8 Figure 6: RK-RBP

P13 $rank(U\otimes V) = l_1 + l_2$ seems to be $l_1 l_2$

Quality
----------
Sec.2.2. argues that the $U \otimes V$ does not cover the entire feature space of bilinear features. I agree that if $U$ and $V$ are fixed, it is correct. However, in deep learning, the dimension reduction and classifier are learned jointly. For Eq.(5), all dimensions in $vec(x_s y_t^T)$ are involved in learning dimension reduction followed by the classifier.

P4. “Because those auxiliary parameters are implicit, we can not train them as other parameters of our model.”, “However, those free auxiliary parameters probably make the learned projection unsuitable.” are based on the author’s speculation.

Novelty
----------
The rank-K projections seem to be novel and effective. In the rank-k projection,  each dimension of the output vector is given by $y_p = {\rm Tr}(U_p^TXV_p )$ and the vector is concatenation of $p=1,..h$ values. In contrast to the conventional rank-1 projection $y = U^T X V$, each dimension of rank-K projection is a summation of k projected values, where $K$ is the rank of $U_p, V_p$. Thus, the rank-K projection is unlikely to miss the discriminative information of the original high-dimensional space when the output dimensions are small.

Reproducibility
--------------------
P7: The number of epochs, batch size, and learning rate for SGD are not shown.



**Strength And Weaknesses:**

Strengths
------------
- The rank-K projection, which performs dimension reduction with a sum of $k$ dimension-reduced vectors, is novel and effective.
- The mathematical discussion seems to be correct.
- This paper shows the connection of the proposed method to existing BiP methods.
- This paper shows the effects of parameters and combination with several improved BP (SMSO, MPN, iSQRT-COV).
- The performance of RK-FBP is higher than state-of-the-art with lower embedding dimensions.

Weakness
--------------
-  Reading this paper is heavy due to many equations.
-  Some discussions are based on the speculation of authors.



**Summary Of The Paper:**

Many Factorized Bilinear Pooling (FBiP) uses Hadamard product-based bilinear projection. This paper reveals that the Hadamard product misses a lot of possible projection directions. This paper proposes a general matrix-based bilinear projection based on rank-k matrix base decomposition. This paper uses the proposed bilinear projection to design RK-FBP. To leverage high-order information in local features, the authors nest several RK-FBP modules.

**Summary Of The Review:**

The Rank-k dimension reduction for bilinear pooling seems novel and effective compared with existing bilinear pooling. It appears to be technically correct. Parameter analysis and comparison with state-of-the-art seem to be sufficient.

---

### Decision · Program_Chairs · 2023-01-20

**Decision:**

Reject

**Justification For Why Not Higher Score:**

The presentation and discussion is not ready yet for publication.

**Justification For Why Not Lower Score:**

N/A

**Metareview: Summary, Strengths And Weaknesses:**

The authors suggest a type of bilinear projection to deal with second order features.  Two reviewers suggest acceptance, one reviewer suggests rejection.

Strengths:
- strong experimental results in comparison to other bilinear pooling techniques for fixed backbones

Weakness:
- paper is very hard to read, there are language issues and
- there is no "proof" for the main result but it is based on claims of the authors
- there are no results where this technique is used for end-to-end training
- the authors manipulate quite heavily the style file leading to the fact that in the original submission (and the revised version) the headers of Section 4 and 4.1 are even overlapping - in the revised version this even worse. The plots and tables are very hard to read.

There were also doubts on the novelty of the approach. While this could not be fully resolved, I think that this paper is not ready yet for publication and the authors should invest considerable effort in improving clarity of the presentation, make their argument precise (ideally with a real proof) and add the requested experiment on end-to-end training.

Details:
- Theorem 1 and the Corollary are straightforward and can be omitted (the corollary is formulated unfortunately, someone could wrongly deduce that the dimension of symmetric matrices is $m^2$ and not $\frac{m(m+1)}{2}$)